# Analysis and prediction of compressive and split-tensile strength of secondary steel fiber reinforced concrete based on RBF fuzzy neural network model

**Song Ling**[1,2]*, **Du Chengbin**[1], **Yao Yafeng**[2,3], **Li Yongheng**[3]

**1** College of Mechanics and Materials, Hohai University, Nanjing, Jiangsu, 210098, China, **2** School of Civil Engineering, Nantong Vocational University, Nantong, 226001, China, **3** Anhui Key Laboratory of Building Structure and Underground Engineering, Anhui Jianzhu University, Hefei, 210037, China

* songlintougao@126.com

**Data Availability Statement:** All relevant data are within the manuscript and its Supporting Information files.

## Abstract

Accurate analysis of the strength of steel-fiber-reinforced concrete (SFRC) is important for ensuring construction quality and safety. Cube compression and splitting tensile tests of steel fiber with different varieties, lengths, and dosages were performed, and the effects of different varieties, lengths, and dosages on the compressive and splitting properties of secondary concrete were obtained. It was determined that the compression and splitting strengths of concrete could be effectively improved by the addition of end-hooked and milled steel fibers. The compressive and splitting strengths of concrete can be enhanced by increasing the fiber length and content. However, concrete also exhibits obvious uncertainty owing to the comprehensive influence of steel fiber variety, fiber length, and fiber content. In order to solve this engineering uncertainty, the traditional RBF neural network is improved by using central value and weight learning strategy especially. On this basis, the RBF fuzzy neural network prediction model of the strength of secondary steel fiber-reinforced concrete was innovatively established with the type, length and content of steel fiber as input information and the compressive strength and splitting tensile strength as output information. In order to further verify the engineering reliability of the prediction model, the compressive strength and splitting tensile strength of steel fiber reinforced concrete with rock anchor beams are predicted by the prediction model. The results show that the convergence rate of the prediction model is increased by 15%, and the error between the predicted value and the measured value is less than 10%, which is more efficient and accurate than the traditional one. Additionally, the improved model algorithm is efficient and reasonable, providing technical support for the safe construction of large-volume steel fiber concrete projects, such as rock anchor beams. The fuzzy random method can also be applied to similar engineering fields.

**Funding:** This work was supported by the Nantong Municipal Science and Technology Program of China [grant number JCZ2022088,SL]; and the Research project of Nantong Vocational University of China[Grant Numbers 22ZK01,YYF].

**Competing interests:** The authors have declared that no competing interests exist.

# 1. Introduction

Compared with primary concrete, secondary concrete is widely used in mass hydraulic concrete because of its lower hydration heat. The crack problem of mass concrete not only affects project quality but also leads to security risks [1–5]. The addition of fibers to a concrete substrate is an important method of improving the brittleness characteristics of concrete and controlling the crack width. Steel-fiber concrete was the earliest application involving the addition of fibers, and thereby, it led to its fastest development [6–10]. Contemporary research, both domestically and internationally, is focused primarily on steel-fiber concrete with smaller coarse aggregates, with regulations in place to support its engineering applications. For instance, the *Technical Regulations for fiber concrete structure* CECS38:2004 to *JG/T472-2015 steel fiber concrete* stipulate that the maximum aggregate size should not exceed 20 mm and 25 mm, respectively [11–13]. Nonetheless, there is a scarcity of research on steel fiber-reinforced concrete with low strength and coarse aggregate sizes exceeding 25 mm. Initial studies indicated that larger aggregates might impede the uniform distribution of fibers, thereby diminishing the material's reinforcing capabilities. Notably, in 1994, Huang Chengkui [14] demonstrated through experimental studies that incorporating 35 mm long steel fibers can result in a secondary steel fiber concrete that matched the original in static, bending, and fatigue strengths. In recent years, Han's research group [15–21] has also confirmed that steel fiber has a certain strengthening and toughening effect on secondary concrete through relevant tests.

As the basic mechanical properties of concrete, compressive strength and splitting tensile strength are considered as the main bases for determining the bearing capacity and cracking resistance of concrete in structural design. Jinrong et al. [22] conducted a test study on the compressive and splitting tensile strengths of aged steel fiber secondary concrete with a bow length of 30, 35, and 60 mm and length of 32 mm. They determined that when compared with primary steel fiber concrete, the modified could not only effectively reduce the amount of cement, adiabatic temperature rise, and dry shrinkage, but also enhanced crack resistance. Simultaneously, it was determined that the ratio of fiber length to aggregate size ($l_f/D_{max}$) was approximately 3/2, steel fiber content was 1.5%, and length of 60-mm splitting tensile strength was the largest. Shi Guozhu's research [23] on 60-mm end-hook steel fiber secondary concrete suggested that longer fibers enhance the toughening effect, with an optimal fiber volume fraction of 1% yielding the best results in terms of cleavage, bending resistance, and toughness. Zhao Mengmeng [24] conducted experiments on steel fiber concrete with various fiber lengths at a volume fraction of 1% and concluded that for C30 concrete, when $l_f/D_{max}$ ratio ranged from 1.25 to 3, there was an effective synergy between steel fiber length and the maximum particle size of the coarse aggregate. This resulted in the splitting tensile strength first increasing and then decreasing with the $l_f/D_{max}$ ratio, with the most significant increase in splitting strength ratio observed. Chen's experiments [25] aligned with Zhao's filitting tensile strength depending on the concrete grade; for C30 steel fiber-reinforced concrete, an increase in fiber length led to reduced strength, whereas for C60 concrete, strength increased with longer fibers. Khaleel Ibrahim & Movahedi Rad [26] studied the plastic properties of beams reinforced by carbon fiber reinforced polymer (CFRP) by using probabilistic design method which takes into account random concrete properties, carbon fiber reinforced polymer (CFRP) properties and complementary strain energy values.The study on the optimal plastic behavior of RC beams strengthened by carbon fiber polymers offers valuable insights into reliability-based design approaches. Despite these studies, there is a dearth of consistent data specifically on the compressive and splitting tensile strengths of secondary steel fiber reinforced concrete. Additionally, the aforementioned research does not fully account for the variability of uncertainty

**Table 1. Mix ratio of secondary steel fiber reinforced concrete.**

| Strength class | Cement ($m_c$) | Water-binder ratio | Sand ratio (%) | Fly ash content (%) | Water reducing agent content (%) | Slump (cm) |
|---|---|---|---|---|---|---|
| C25 | 276 | 0.43 | 44 | 18 | 1.3 | 5–9 |

in strength distribution that might occur in actual engineering applications, which could lead to discrepancies in results and potentially compromise safety.

In view of the uncertainty distribution of steel fiber reinforced concrete strength in practical engineering, some scholars try to use intelligent algorithms to make comprehensive analysis in order to improve efficiency. Oveys et al. [27] presents an investigation into the bond strength of travertine, granite, and marble, to a concrete substrate using a shear-splitting test. Based on the findings, a novel fuzzy logic approach was proposed to predict the bond strength. Wang et al. [28] established random forest (RF) to predict UCS by analyzing and comparing five traditional models: RF, multiple regression analysis (MR), backpropagation neural network (BPNN), extreme learning Machine (ELM) and support vector regression (SVR). Pouria et al. [29] compared traditional backpropagation algorithms (LM), differential evolution (DE), and particle swarm optimization (PSO). On this basis, artificial neural network (ANN) technology is combined with a robust optimization technique PSOTD to predict the CS of RHA concrete. Through the analysis of these documents, it is found that most of the current intelligent algorithm models of engineering prediction only focus on randomness or fuzziness of engineering, and do not consider the two comprehensively. This will cause results to deviate from reality. Therefore, in view of the influence of different types, lengths and quantities of steel fibers on the performance of secondary steel fiber reinforced concrete, the traditional RBF neural network is improved, and the optimized fuzzy RBF neural network is established to be a more effective tool for the performance prediction of steel fiber reinforced concrete.

## 2. Materials

In the experimental setup, C25 concrete-often designated for hydraulic structures—served as the matrix concrete. The mix retained consistent ratios with standard non-fiber-reinforced concrete, substituting large aggregates partially with steel fibers and other components. The details of the mix design are presented in Table 1. The materials utilized for the concrete mix included P.O42.5 ordinary Portland cement, natural river sand with a fineness modulus of 2.65, and coarse aggregate sizes ranging from 5 to 20 mm and 20 to 40 mm, in a mass ratio of 4:6. The supplementary cementitious material was grade I fly ash, and the chemical admixture used was PCA-I type water-reducing agent, with the water sourced from the Mechanics laboratory taps at Hohai University. Three different shapes of steel fibers—shear, end hook, and milled Harix-type—were sourced from Hebei Zhitai Steel Fiber Company. The specific details and experimental conditions related to the fibers are documented in Tables 2 and 3, and their geometrical shapes are illustrated in Figs 1 and 2. Nine sets of concrete specimens were

**Table 2. Fiber parameters and dosage.**

| Fiber type | Average length (mm) | Equivalent diameter (mm) | Slenderness ratio | Tensile strength N ($mm^2$) | Volume fraction (%) |
|---|---|---|---|---|---|
| End-hooked | 60 | 0.75 | 80 | 1150 | 0/0.5/1.0/1.5/2 |
| | 50 | 0.75 | 65 | 1150 | 1.0 |
| | 35 | 0.55 | 65 | 1250 | 1.0 |
| Shear type | 38 | 1.4 | 27 | 800 | 1.0 |
| Milling type | 32 | 0.9 | 35 | 700 | 1.0 |

**Table 3. Mix ratio of secondary steel fiber reinforced concrete.**

| ID | Water-binder ratio | Sand ratio(%) | Dosage of concrete materials (kg/m$^3$) | | | | | |
|----|--------------------|---------------|------------|-------|--------|------|----------------------------------|----------------------------------|
|    |                    |               | Steel fiber | Water | Cement | Sand | Particle size stone(5~20mm) | Particle size stone(20~40mm) |
| F0 | 0.43 | 44 | 0 | 145 | 276 | 832 | 424 | 636 |
| J310 | 0.43 | 44 | 78.5 | 145 | 276 | 811 | 424 | 609 |
| X310 | 0.43 | 44 | 78.5 | 145 | 276 | 811 | 424 | 609 |
| D310 | 0.43 | 44 | 78.5 | 145 | 276 | 811 | 424 | 609 |
| D510 | 0.43 | 44 | 78.5 | 145 | 276 | 811 | 424 | 609 |
| D605 | 0.43 | 44 | 39.25 | 145 | 276 | 822 | 424 | 623 |
| D610 | 0.43 | 44 | 78.5 | 145 | 276 | 811 | 424 | 609 |
| D615 | 0.43 | 44 | 117.75 | 145 | 276 | 801 | 424 | 596 |
| D620 | 0.43 | 44 | 157 | 145 | 276 | 790 | 424 | 582 |

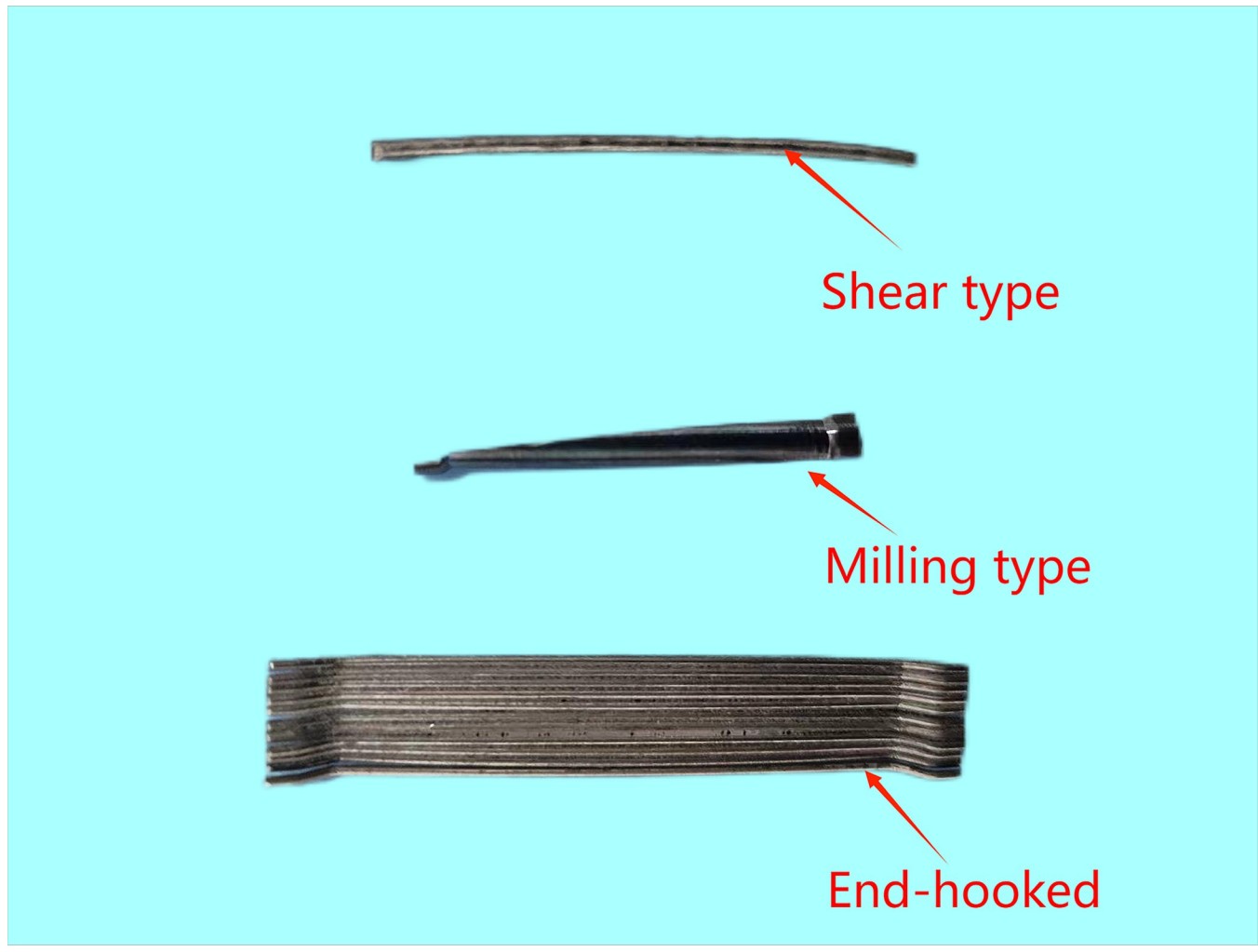

**Fig 1. Physical diagram of three different types of steel fibers.**

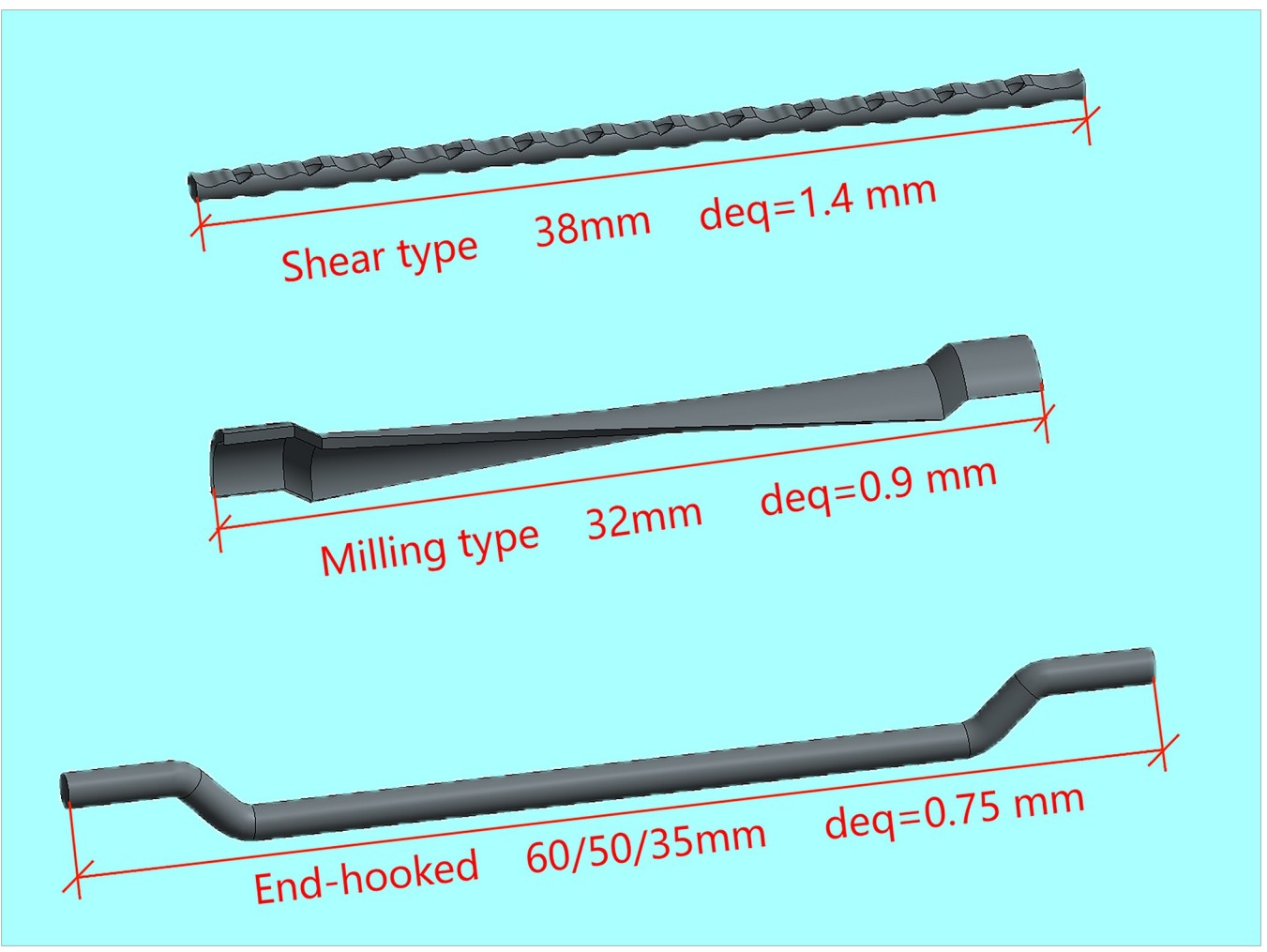

**Fig 2. Geometry of three different types of steel fibers.**

prepared for testing. The mechanical properties of interest, namely cube compressive strength and splitting tensile strength, were assessed using standard 150 mm cubic specimens, with six replicates per set. The preparation process involved mixing the components in a forced-action mixer, consolidating the mix on a vibrating table, demoulding after 24 hours, and subsequently curing the specimens in a controlled environment for 28 days. For the testing phase, the procedures aligned with the GB-T5008-2019 *Standard for Testing Methods of Mechanical Properties of Ordinary Concrete*. The application of load to the specimens is conducted using an electro-hydraulic servo universal testing machine. The compressive loading device is shown in Fig 3, and the split-pull loading device is shown in Fig 4.

For specimen preparation, all components except the steel fibers were initially combined in a forced mixer, where they underwent wet mixing for one minute. Subsequently, steel fibers were introduced, and the mixing continued for an additional 2 min. The resultant concrete mix is depicted in Fig 5, showcasing a uniform distribution of steel fibers. The mix demonstrated favourable cohesion and water retention properties. However, a minor degree of fiber clumping was observed when the fiber content reached 2%. This mixture was then poured into plastic moulds for shaping. The compaction process involved the use of a vibrating table, which ensured the mix was densely packed and free of voids.

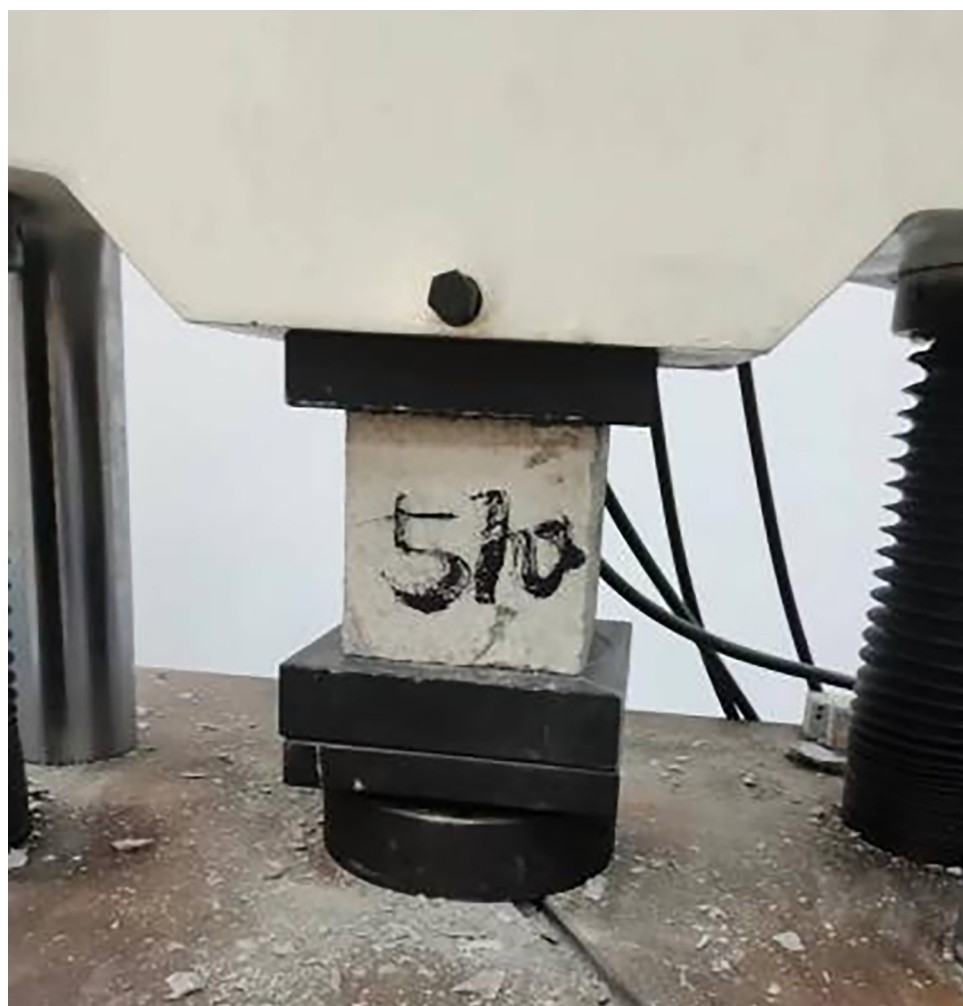

**Fig 3. Compressive loading device.**

After 28 days of curing, take out the steel fiber concrete specimen, wipe the specimen clean and check the flatness and perpendicularity of the specimen. Taking the side of the specimen as the bearing surface, the concrete cube compression test was carried out. A steel plate was added between the test plate and the specimen, and a steel ball seat was placed between the lower plate and the steel plate. Set the loading speed of the testing machine to 0.4MPa /s, and start the testing machine for testing. When the upper pressure plate was in contact with the specimen, adjust the ball seat to make the specimen under uniform pressure. The whole test process was automatically loaded by the host machine, and the failure load was recorded when the specimen was broken. The test is precisely set to 0.01KN.

The splitting tensile test was carried out by drawing parallel positioning lines at the center of the two opposite sides of the specimen. Place the fixture in the center of the lower clamping plate of the testing machine. Then the specimen was centered in the fixture, and finally the pad was placed in the position of the positioning line of the upper and lower pressure surface of the specimen. Set the loading speed of the testing machine to 0.035MPa/s, and start the testing machine to carry out the splitting tensile test. When the upper pressure plate was close to the specimen, adjust the ball seat to make the specimen under uniform pressure. The whole test

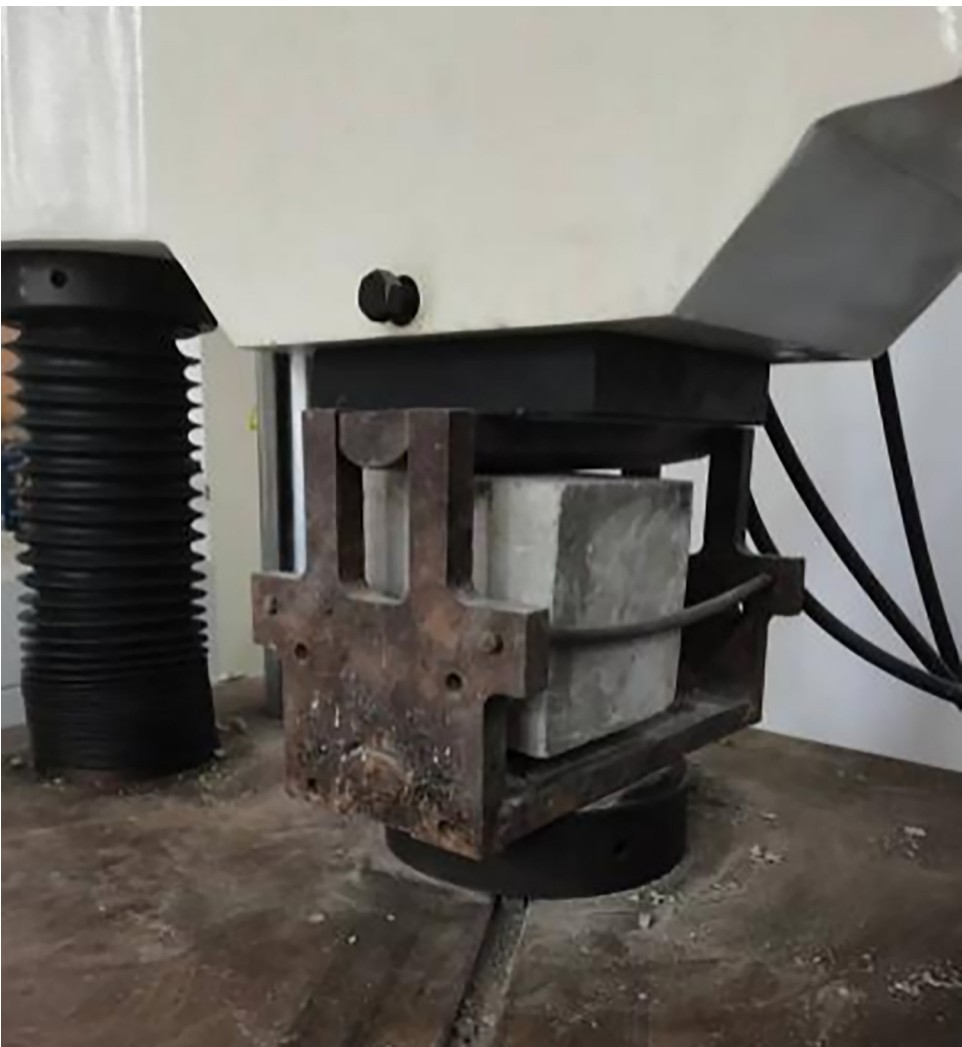

**Fig 4. Split-pull loading device.**

process was automatically loaded by the host machine, and the failure load was recorded when the specimen was broken. The test is precisely set to 0.01KN.

## 3. Test methods and analysis

### 3.1. Routine analysis

In line with the data processing guidelines stipulated by the GB-T5008-2019 *Standard for Ordinary Concrete Mechanical Properties Test Method*, if one of the three recorded values—either the maximum or minimum—deviates by more than 15% from the median, the maximum and minimum values are discarded. The remaining median value is then adopted as the representative strength value for that set of specimens. The calculations for compressive strength, splitting tensile strength, and strength gain are conducted according to the protocols detailed in Table 4.

From the data presented in Table 4, it is evident that the addition of steel fibers leads to a modest enhancement in the compressive strength of concrete cubes, with an increase of up to

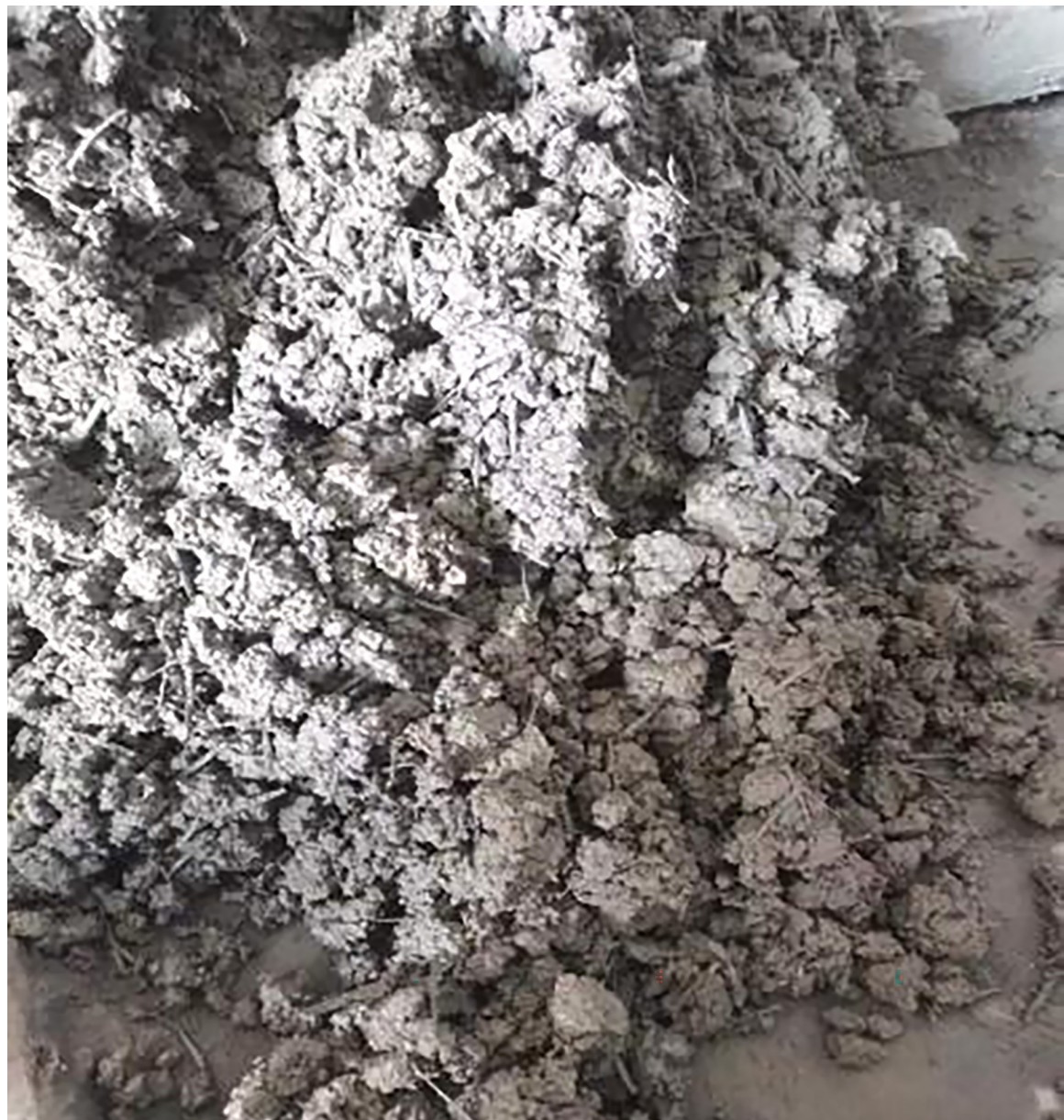

**Fig 5. Steel fiber reinforced concrete mix.**

approximately 30%. However, the impact on splitting tensile strength is much more pronounced, with improvements reaching up to 95%. This significant difference is largely due to the nature of the tensile test, which involves pulling out the fibers, necessitating a greater load to achieve failure. Despite the relatively modest gains in compressive strength with steel fiber reinforcement, the failure mode of the concrete is significantly distinct from that of non-fiber-reinforced matrix concrete. After reaching the peak load, steel fiber-reinforced concrete is capable of bearing additional load, demonstrating a form of post-crack load-bearing capacity. Conversely, matrix concrete without fibers typically exhibits a sudden and brittle failure upon reaching its maximum load.

**Table 4. Analysis of compressive and splitting tensile strength test results of specimens.**

| Different working conditions | Average compressive strength $f_c^{SF}$ (MPa) | Compressive strength gain ratio $f_c^{SF}/fc$ | Average splitting tensile strength $f_t^{SF}$ (MPa) | Split tensile strength gain ratio $f_t^{SF}/ft$ |
|---|---|---|---|---|
| F0 | 30.3 | 1 | 2.41 | 1 |
| J310 | 28.4 | 0.94 | 2.36 | 0.98 |
| X310 | 38.4 | 1.27 | 3.15 | 1.31 |
| D310 | 35.7 | 1.18 | 3.58 | 1.49 |
| D510 | 39.3 | 1.30 | 4.57 | 1.90 |
| D605 | 35.0 | 1.16 | 3.17 | 1.32 |
| D610 | 38.0 | 1.26 | 4.10 | 1.71 |
| D615 | 39.5 | 1.31 | 4.68 | 1.95 |
| D620 | 37.5 | 1.24 | 3.92 | 1.63 |

**3.1.1. Analysis of cube compressive strength test results.** Figs 6–8 illustrate the comparative analysis of the cubic compressive strength ($f_{fc}$) of steel fiber-reinforced concrete, highlighting the effects of varying the type of steel fibers, their lengths, and their volume fractions, against the compressive strength of ordinary concrete. As depicted in Fig 6, the cube compressive strength for all three steel fiber-reinforced concretes, with fibers approximately 35 mm in length and at a 1% volume fraction, shows an increase in strength compared to the matrix concrete. Notably, the milled steel fiber-reinforced concrete exhibits the most substantial enhancement, with a 27% increase as detailed in Table 5. This improvement is attributed to the unique shape of the milled fibers—radial distortion with hooks and anchor tails at both ends—facilitating effective stress transfer under compression. However, the shear steel fibers, characterized by their larger diameter and lower slenderness ratio, contribute less to the compressive strength than the matrix concrete. This outcome suggests that shear steel fibers may not be optimal for concrete components designed primarily to withstand compressive forces. Fig 7 reveals that the compressive strength of concrete cubes reinforced with 1% volume fraction of end-hook steel fibers, regardless of their length, was superior to that of the matrix concrete. This indicates a consistent beneficial effect of end-hook steel fibers on the compressive strength of the concrete cubes. As shown in Fig 8, when the 60mm end-hook steel fiber reinforced concrete content is 1.5%, the strengthening effect is the best. It can be seen that the steel fiber has limited effect on the compressive strength of concrete cube. Therefore, it is not recommended to use high-content steel fiber to improve the compressive strength.

**3.1.2. Analysis of splitting tensile strength results.** Figs 9–11 show the relationship between the splitting tensile strength ($f_{ft}$) of steel fiber-reinforced concrete and splitting tensile strength of ordinary concrete matrix with changes in the steel fiber variety, length, and content. Fig 9 shows that, when compared to the matrix concrete, the inclusion of steel fibers of similar lengths at a uniform content level generally enhances the splitting tensile strength across all fiber types except for shear steel fiber-reinforced concrete. Notably, the shear steel fibers yield the most significant improvement, with a 49% increase in strength. Meanwhile, as indicated in Fig 10, the splitting tensile strength for concrete with 1% volume of end-hook steel fibers of various lengths is considerably higher than that of the matrix concrete. The peak tensile strength is observed with fibers measuring 50 mm, registering a 90% increase, which surpasses the performance of the D610 standard. The test findings suggest that the optimal $l_f/D_{max}$ ratio, where lf is the fiber length and $D_{max}$ is the maximum aggregate size, is 1.25. Fig 11 further illustrates that for 60-mm end-hook steel fiber-reinforced concrete at a 1.5% fiber volume fraction, the splitting tensile strength reaches its maximum, with the increase rate at 95%.

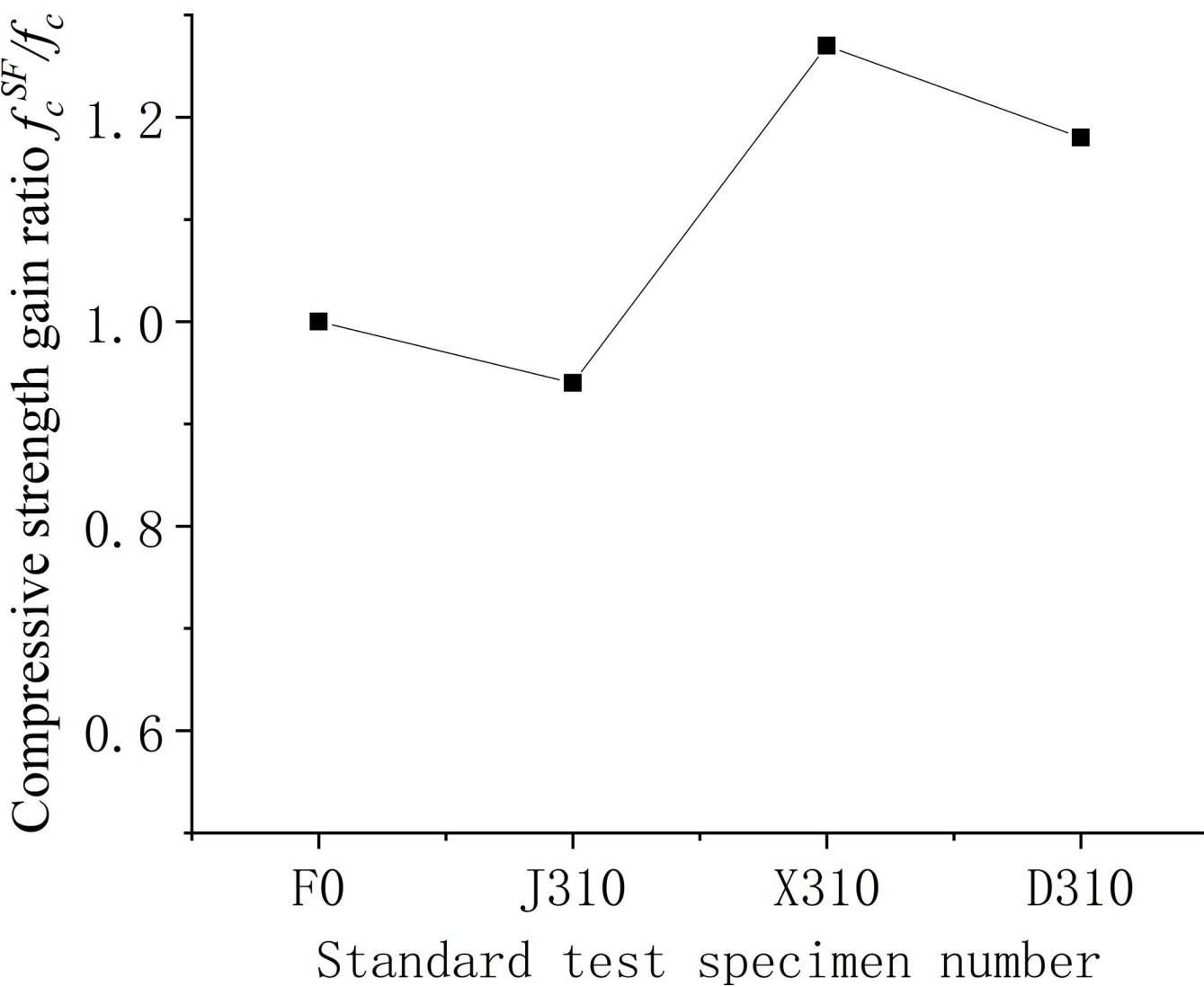

**Fig 6. Relationship between compressive strength and fiber varieties of concrete cube.**

However, at a 2% volume fraction, the incidence of larger unvibrated and uncompacted voids increases, leading to poor encapsulation of fibers by the cement paste, which detrimentally affects the splitting tensile strength. Comprehensive analysis of compressive and splitting tensile strengths suggests that the optimal fiber length for low-strength secondary steel fiber-reinforced concrete is 50 mm, with a content of 1%. At this specification, the strengthening effect is observed to be most beneficial. It is advised against using high-volume fractions of steel fibers as this can lead to clumping issues.

Fig 12 presents the regression analysis results for the splitting tensile strength of three different end-hook steel fiber-reinforced concretes. The analysis highlighted that when the volume fraction of 60-mm long steel fibers reached 2%, there was a notable formation of fiber clusters, which led to a reduction in strength. Consequently, to enhance the accuracy of the statistical analysis, data pertaining to the D620 specimen were excluded. The regression yielded an influence coefficient ($\alpha_t$) of 0.85 for the splitting tensile strength of end-hook steel fiber-reinforced concrete. This coefficient is a measure of the effect that the end-hook steel fibers

**Table 5. Compressive strength test results of specimens.**

| Different working conditions | Sample 1 $f_c^{SF}$ (MPa) | Sample 2 $f_c^{SF}$ (MPa) | Sample 3 $f_c^{SF}$ (MPa) |
|---|---|---|---|
| F0 | 33.7 | 24.9 | 30.3 |
| J310 | 28.5 | 28.4 | 28.2 |
| X310 | 37.7 | 39.2 | 38.2 |
| D310 | 38.2 | 36.4 | 32.5 |
| D510 | 40.2 | 40.5 | 37.2 |
| D605 | 32.9 | 37.3 | 34.7 |
| D610 | 38.1 | 38.2 | 37.8 |
| D615 | 40.7 | 41.0 | 36.9 |
| D620 | 40.0 | 33.7 | 38.7 |

Note: The specimen number F0 indicates the matrix concrete without fiber; D, X, and J denote the end-hook type, milling type, and shearing type, respectively; for example, D605 indicates that the end-hook-type fiber length is 60 mm and the volume fraction is 0.5% of the specimen; and D310 indicates that the end-hook-type fiber length is 35 mm and volume fraction is 1.0% of the specimen.

have on the tensile strength of the concrete. Notably, the calculated $\alpha_t$ value of 0.85 exceeds the influence coefficient of 0.76 recommended by the specification *JG-T472-2015 Steel Fiber Reinforced Concrete*, where the maximum particle size of coarse aggregate is limited to 25 mm. The implication of these findings is that the tensile strength and toughness of low-strength secondary concrete can be significantly improved through the addition of end-hook steel fibers. Furthermore, the actual test values for tensile strength exceeded those of the specified standard values, suggesting that end-hook steel fibers are particularly effective in enhancing these properties in concrete.

## 3.2. Uncertainty analysis

**3.2.1. Uncertainty analysis of cube compressive strength results.** The compressive strength is calculated according to Formula (1) of the code:

$$f_c^{SF} = \frac{F}{A} \tag{1}$$

where $f_c^{SF}$ denotes the compressive strength of the steel-fiber concrete; $F$ denotes the failure load of the specimen, and $A$ denotes the bearing area of the specimen. The calculated results for each specimen group are listed in Table 5.

Figs 13–15 depicts the variability in compressive strength across three types of fiber-reinforced concrete specimens. In Fig 13, a comparison between different fiber types within the concrete matrix highlights the considerable variability in compressive strength of the non-fiber-reinforced matrix concrete. This variability is attributed to the inconsistent dispersion of large-particle aggregates within the matrix. In contrast, the milled steel fiber-reinforced concrete, which features shorter, flakier fibers, displays a more uniform dispersion, resulting in lower variability and more reliable compressive strength values. Fig 14 compares the compressive strengths of end-hook fiber-reinforced concrete of varying lengths to that of the matrix concrete. The results indicate that the addition of fibers tends to reduce the uncertainty in compressive strength measurements, suggesting a more consistent behaviour of the material under compressive loads. Lastly, Fig 15 demonstrates that there is a higher degree of uncertainty in the compressive strength when the fiber content is either too low or too high.

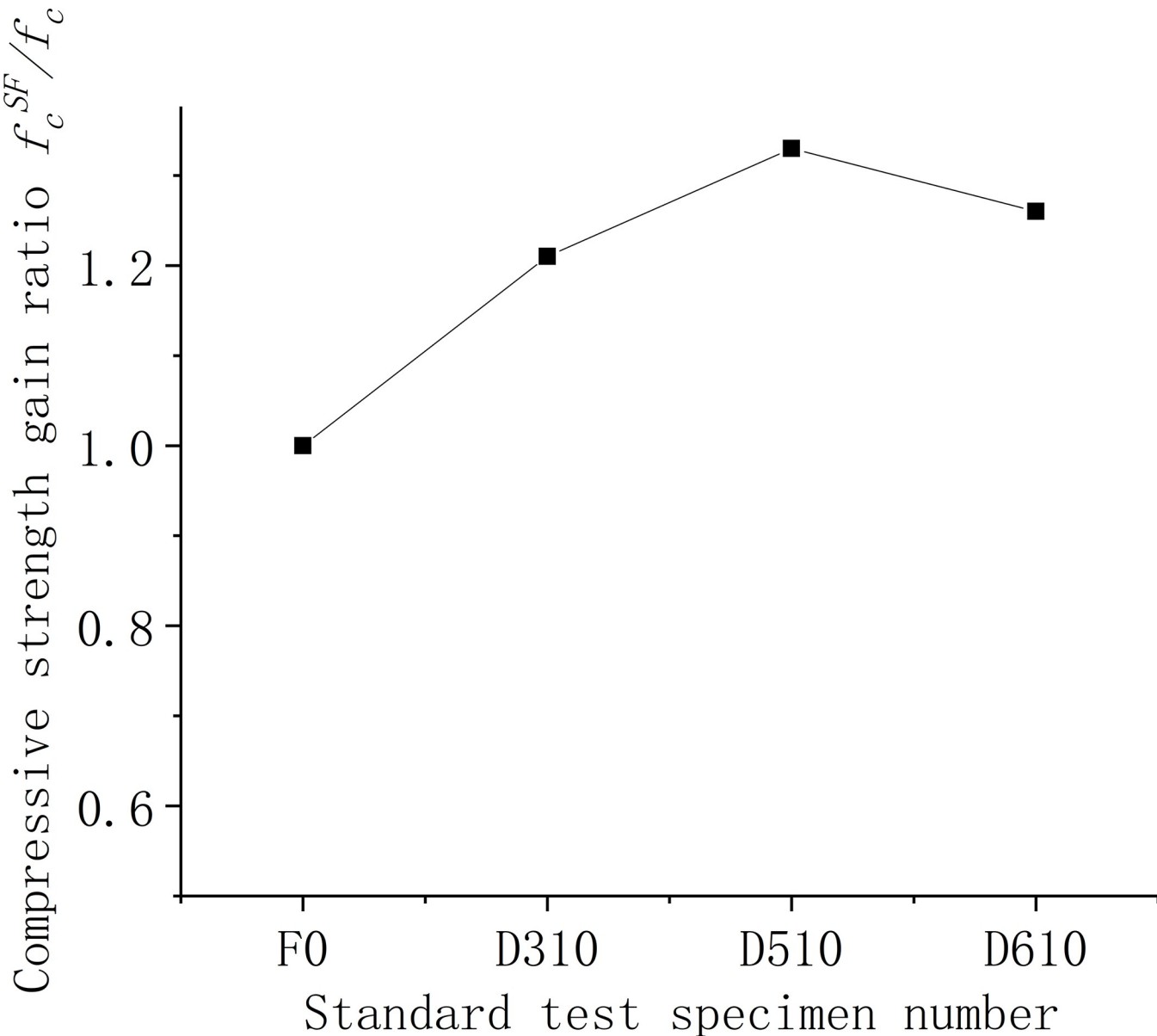

**Fig 7. Relationship between compressive strength and fiber length of concrete cube.**

**3.2.2. Uncertainty analysis of splitting tensile strength results.** The splitting tensile strength is calculated according to the standard Formula (2).

$$f_t^{SF} = \frac{2F}{\pi A} = 0.637\frac{F}{A} \tag{2}$$

where $f_t^{SF}$ denotes the splitting tensile strength of the steel fiber-reinforced concrete, $F$ denotes the failure load of the specimen, $A$ denotes the area of the splitting surface of the specimen. The calculated results for each specimen group are listed in Table 6.

Figs 16–18 displays the splitting tensile strengths for three types of fiber-reinforced concrete samples. In Fig 16, the comparison between various fiber-reinforced concretes and the baseline

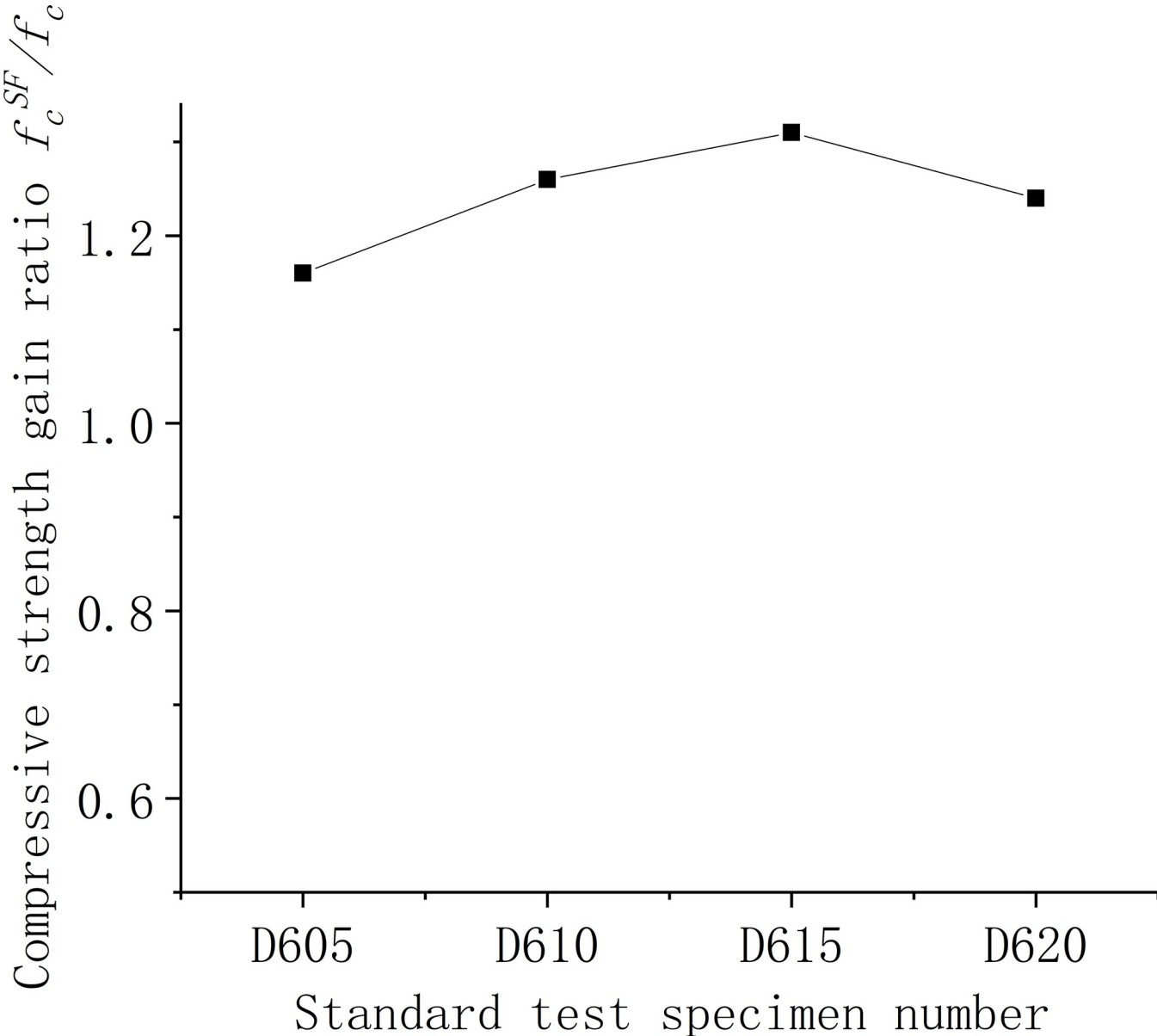

**Fig 8. Relationship between compressive strength and fiber content of concrete cube.**

matrix concrete demonstrates a pronounced variability in splitting tensile strength for the matrix concrete lacking fibers. The introduction of fibers is shown to mitigate this variability, leading to a more uniform distribution of strength values. Fig 17 examines the relationship between fiber length in end-hook fiber concrete and splitting tensile strength, relative to the standard matrix concrete. It is apparent that increasing the length of the fibers tends to decrease the uncertainty in splitting tensile strength, implying that longer fibers contribute to a more consistent tensile response in the concrete. Lastly, Fig 18 indicates that both low and high fiber content levels correspond with greater uncertainty in splitting tensile strength.

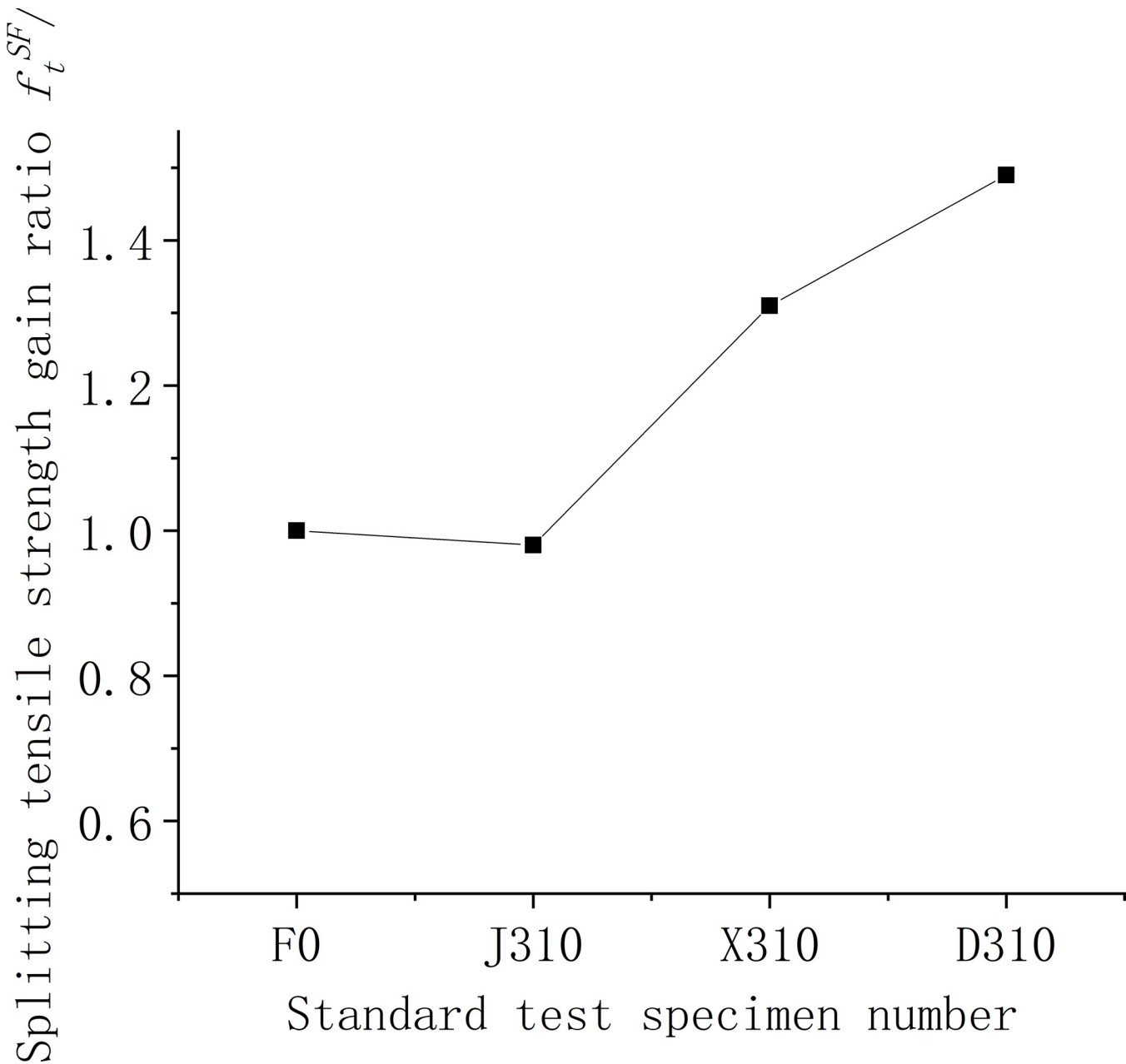

**Fig 9. Relationship between splitting tensile strength and fiber varieties of concrete cube.**

## 4. RBF fuzzy neural network prediction model for strength of secondary steel fiber reinforced concrete

Considering the variable nature of the mechanical properties in steel fiber-reinforced concrete, solely depending on experimental data fitting is inadequate for providing precise and comprehensive compressive and splitting tensile strength values for practical engineering applications. Consequently, this study employs an enhanced artificial intelligence algorithm model to conduct the inaugural predictions of compressive and splitting tensile strength for steel fiber-reinforced concrete under uncertain engineering conditions [30].

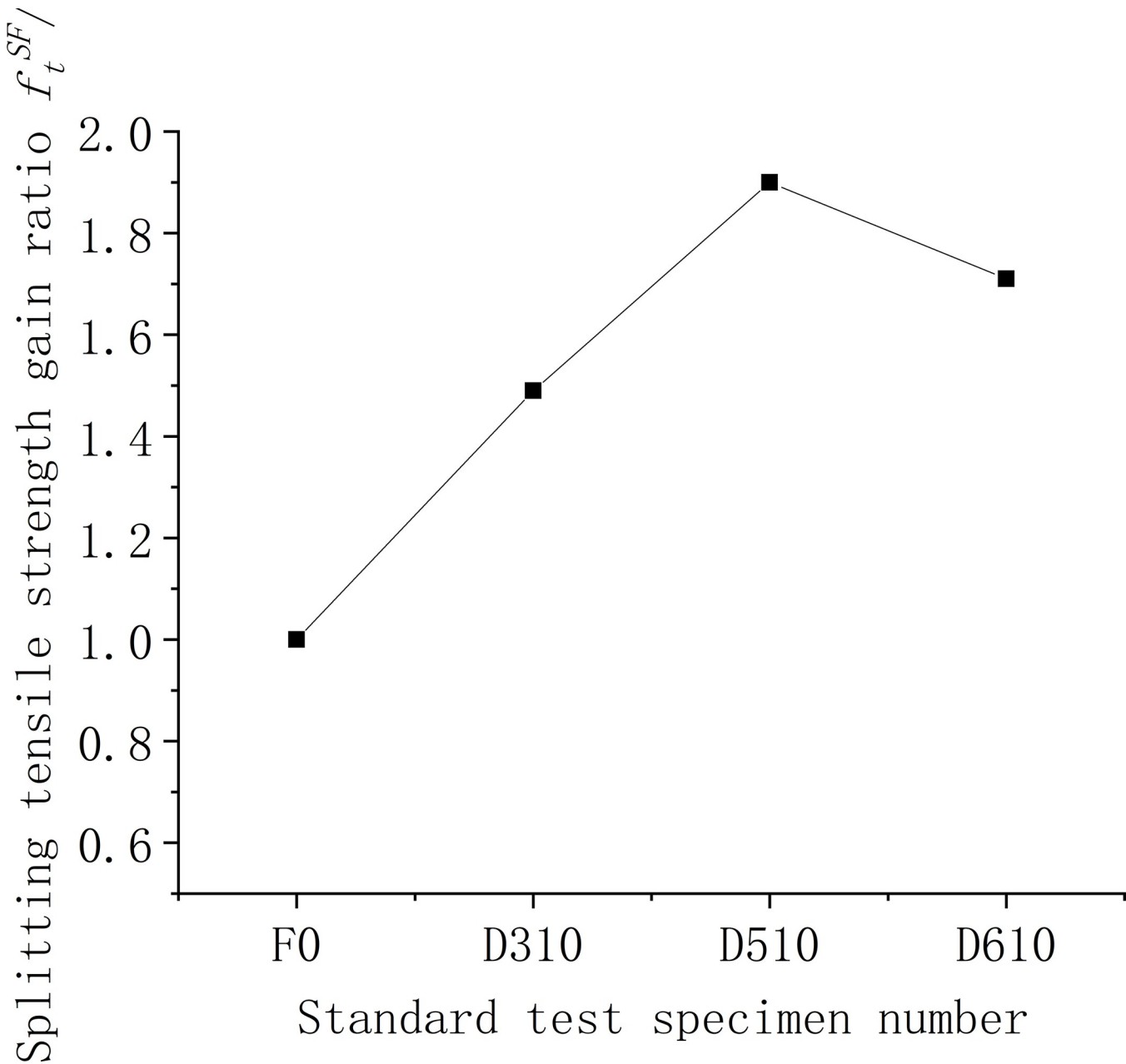

**Fig 10. Relationship between splitting tensile strength and fiber length of concrete cube.**

## 4.1. RBF model and improvement

In 1988, Bromhead and Lowe applied the RBF into neural network models. Through iterative interpolation, they established a structured three-layer topology consisting of an input layer, a hidden layer, and an output layer. The RBF neural network model is distinguished by its organized architecture, efficient training processes, and robust convergence properties, making it extensively utilized in addressing nonlinear engineering challenges [31]. The architecture of the model is depicted in Fig 19.

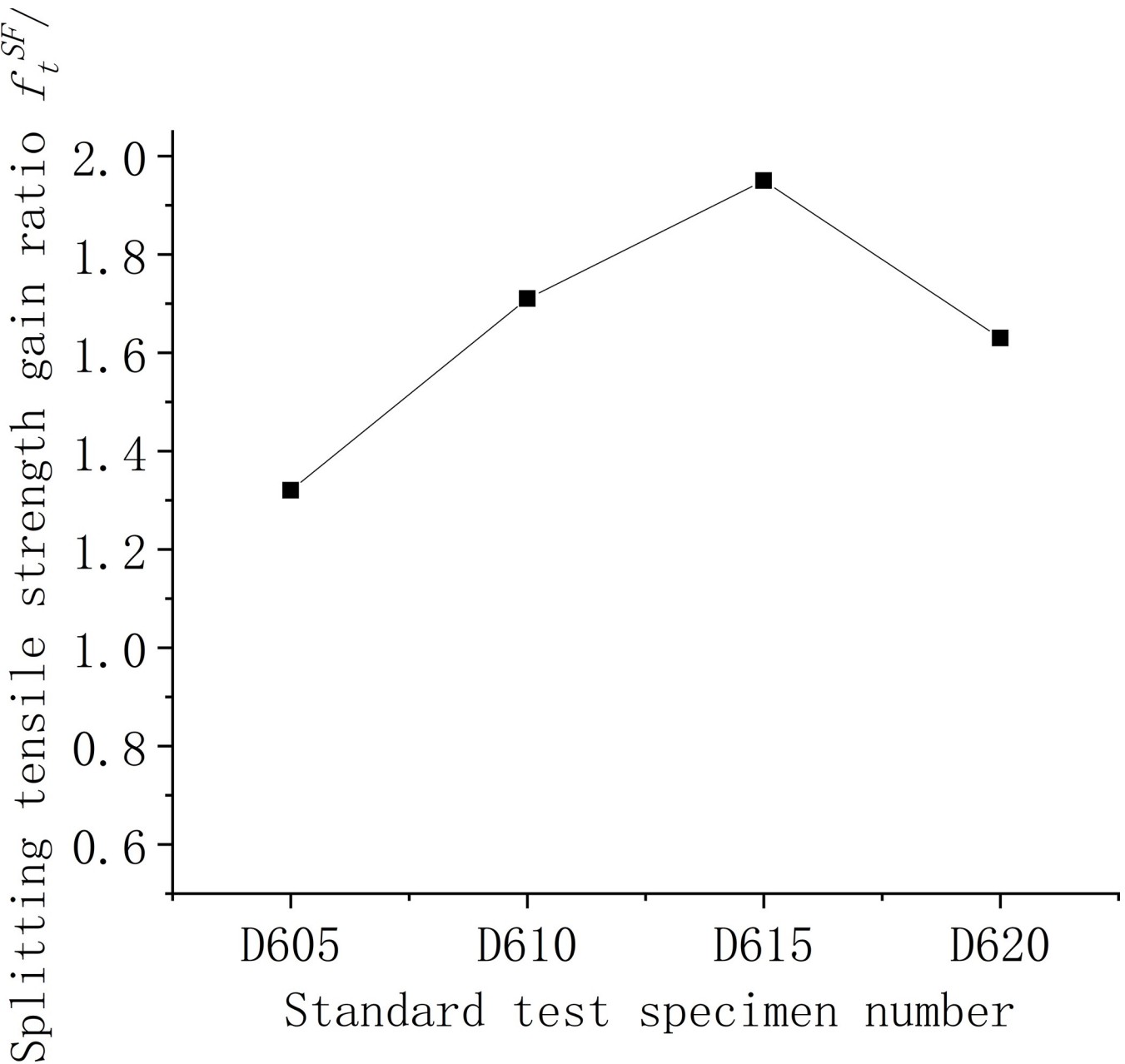

**Fig 11. Relationship between splitting tensile strength and fiber content of concrete cube.**

When traditional RBF neural networks solve engineering problems, the numbers of input and output parameters are not strictly set. For example, for RBF neural networks with multiple inputs and single outputs, the output function can be expressed as follows:

$$y = \sum_{k=1}^{K} \omega_k \varphi_k(\mathrm{xi}) \tag{3}$$

where $Xi$ represents each input variable of the network, $K$ denotes the number of neurons in the hidden layer, $\omega_k$ represents the interpolation weight of the $KTH$ neuron, and $\varphi_k$ denotes

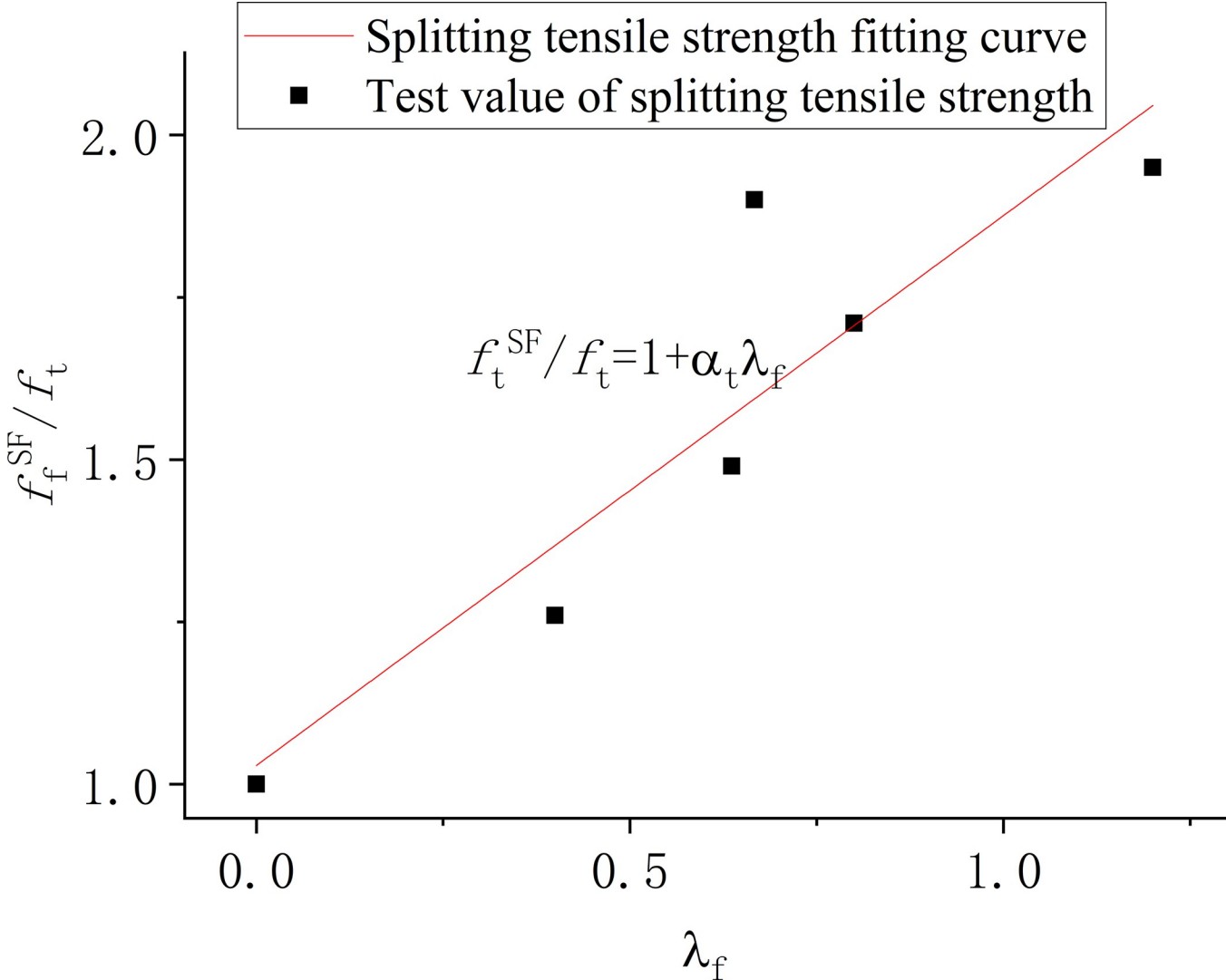

**Fig 12. Statistical analysis of the splitting tensile strength of steel fiber reinforced concrete.**

the corresponding neuron output value. Based on the radial homogeneity principle of the interpolation distance,

$$\varphi_k(x) = \exp(-\|x - \mu_k\|/\sigma_k^2) \tag{4}$$

where $\sigma_k$ and $\mu_k$ denote the training variance and central values of the network model, respectively. Therefore, by combining the above two formulae, the total output expression of the RBF neural network can be obtained as follows:

$$y = \sum_{k=1}^{K} \omega_k \exp(-\|x - \mu_k\|/\sigma_k^2) \tag{5}$$

A traditional RBF neural network uses the interpolation distance of the radial basis function to iterate the final output value and achieve certain results in solving general mathematical problems. However, in view of uncertainties, such as force field distribution and material

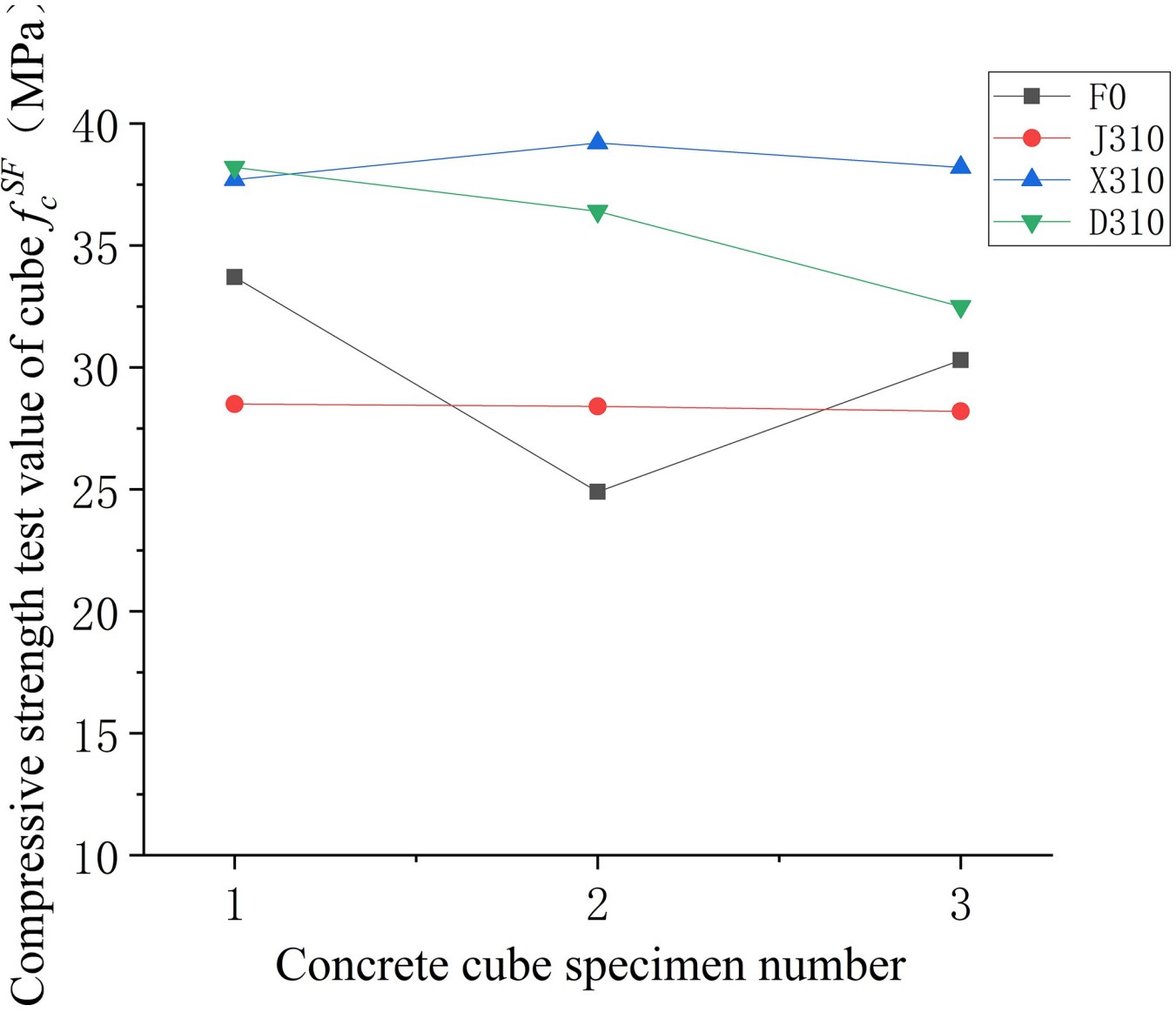

**Fig 13. Compressive strength values of concrete with different fibre varieties.**

uniformity in practical engineering problems, traditional radial basis function interpolation is powerless [32,33]. Therefore, to adapt to this uncertainty, the traditional RBF neural network must be improved as follows:

(1) Fuzzy improvement of central value learning strategy

Through fuzzy clustering analysis, the nonuniform coefficient is fused, and central value learning is decomposed into several different parts for simultaneous processing to optimise the Euclidean shortest distance generated during the radial basis function iteration as follows:

$$\tilde{c}(x) = \mathrm{argmin}\tilde{\rho}_k \|x(n) - c_k(n)\|, k = 1, 2, \cdots m \tag{6}$$

Where $\tilde{\rho}_k$ denotes the non-uniform coefficient, and $\tilde{c}(x)$ denotes the fuzzy matching centre of the input.

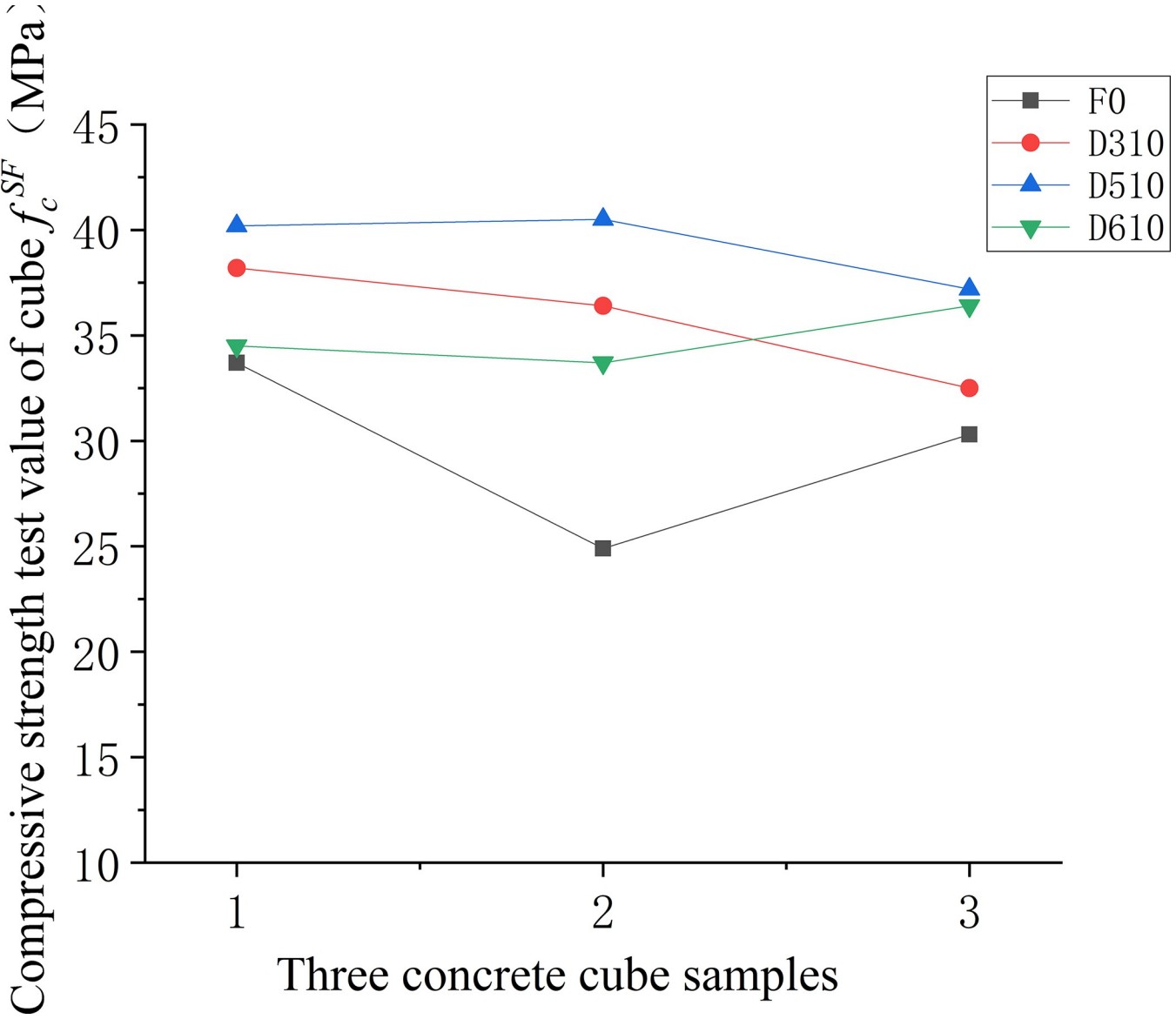

**Fig 14. Compressive strength values of concrete with different fibre lengths.**

Similarly, the policy centre of the radial basis function should be optimised as follows:

$$\tilde{c}_k(n+1) = \begin{cases} \tilde{c}_k(n) + \eta\tilde{\rho}_k[x(n) - \tilde{c}_k(n)], \tilde{c}_k = \tilde{c}(x) \\ \tilde{c}_k(n) \qquad\qquad\qquad\quad , \tilde{c}_k \neq \tilde{c}(x) \end{cases} \qquad (7)$$

(2) Fuzzy improvement of weight learning strategy

Through the gradient descent method, a random forgetting factor was introduced to improve the training and learning strategies based on the principle of minimising the output function value to fuzzy-adjust the output weight of the original model. The original objective

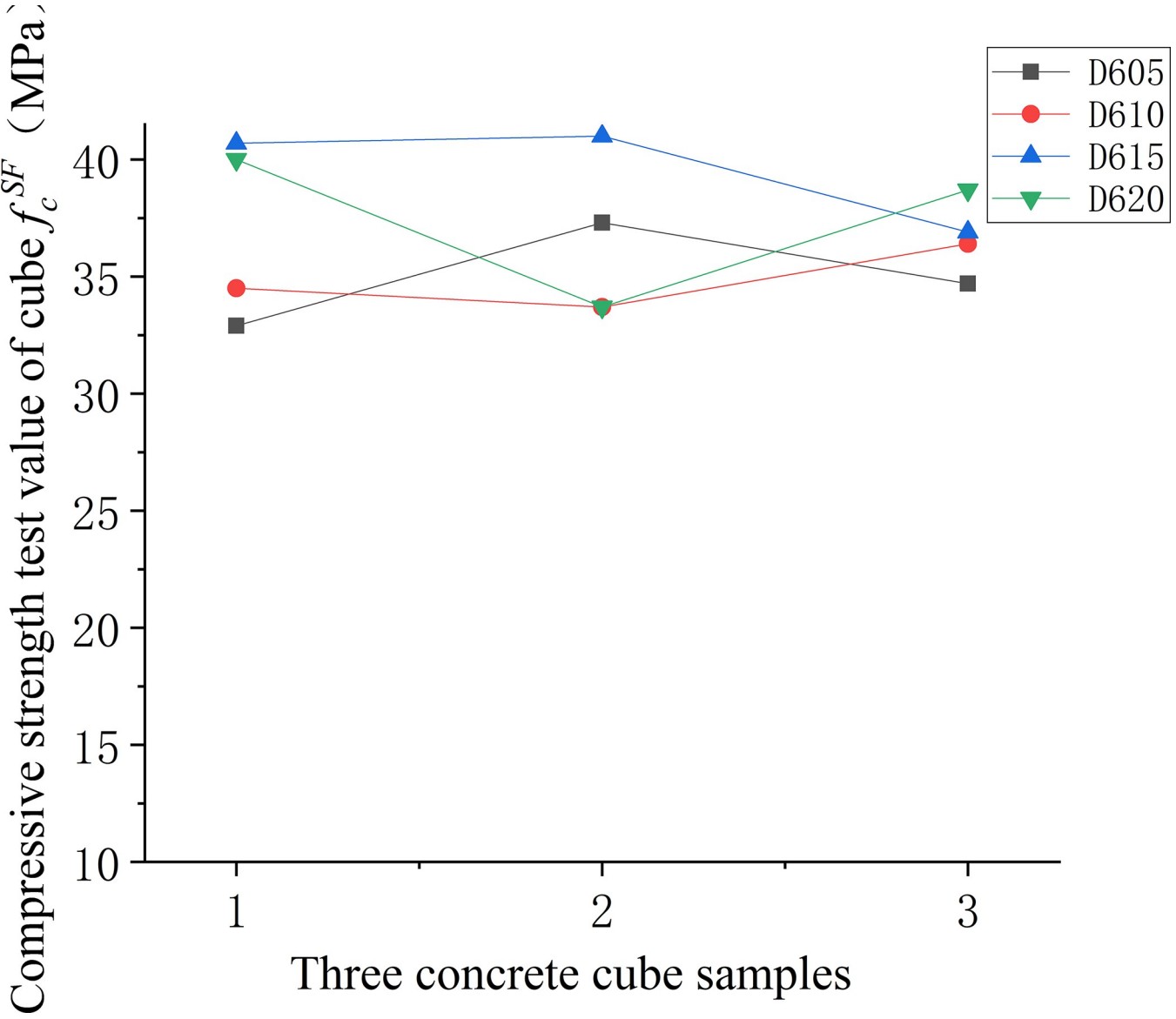

**Fig 15. Compressive strength values of concrete with different fibre content.**

function can be modified as follows:

$$minE = \frac{1}{2}\sum_{j=1}^{N}\beta_j e_j^2 \tag{8}$$

Where $\beta_j$ denotes the random forgetting factor and $e_j$ denotes the error signal. Furthermore, the shortest Euclidean distance generated by the iteration of the radial basis function can be expressed as follows:

$$e_j = y_j - \sum_{i=1}^{k}\tilde{\omega}_i \exp(-\|x - \mu_i\|/\sigma_i^2) \tag{9}$$

**Table 6. Test results of splitting tensile strength of specimens.**

| Different working conditions | Sample 1 $f_t^{SF}$ (MPa) | Sample 2 $f_t^{SF}$ (MPa) | Sample 3 $f_t^{SF}$ (MPa) |
|---|---|---|---|
| F0 | 2.62 | 1.94 | 2.41 |
| J310 | 2.53 | 2.26 | 2.28 |
| X310 | 2.84 | 3.23 | 3.37 |
| D310 | 3.76 | 3.43 | 3.54 |
| D510 | 4.67 | 4.57 | 4.47 |
| D605 | 3.57 | 3.04 | 2.90 |
| D610 | 3.75 | 4.38 | 4.18 |
| D615 | 4.48 | 4.79 | 4.78 |
| D620 | 3.60 | 4.01 | 4.13 |

According to the gradient descent algorithm and RBF neural network radial basis function iterative steps, the gradient function of the width of the neurons in each hidden layer $\delta_i$, radial basis function centre $c_i$, and output weight $\omega_i$ can be expressed as follows:

$$\nabla_{\delta_i} F(x) = \frac{2\omega_i}{\delta_i^3} \varphi_i(\|x_j - c_i\|)\|x_j - c_i\|^2 \tag{10}$$

$$\nabla_{c_i} F(x) = \frac{2\omega_i}{\delta_i^2} \varphi_i(\|x_j - c_i\|)\|x_j - c_i\| \tag{11}$$

$$\nabla_{\omega_i} F(x) = \varphi_i(\|x_j - c_i\|) \tag{12}$$

Aiming at the uncertainty of the project, a random forgetting factor and nonuniform coefficient are introduced to improve the values of $\delta_i$, $c_i$, and as follows:

$$\Delta\tilde{\delta}_i = \eta\tilde{\rho}_i \frac{2\tilde{\omega}_i}{\delta_i^3} \sum_{j=1}^{N} \beta_j e_j \varphi_i(\|x_j - c_i\|)\|x_j - c_i\|^2 \tag{13}$$

$$\Delta\tilde{c}_i = \eta\tilde{\rho}_i \frac{2\tilde{\omega}_i}{\delta_i^2} \sum_{j=1}^{N} \beta_j e_j \varphi_i(\|x_j - c_i\|)\|x_j - c_i\| \tag{14}$$

$$\Delta\tilde{\omega}_i = \eta\tilde{\rho}_i \sum_{j=1}^{N} \beta_j e_j \varphi_i(\|x_j - c_i\|) \tag{15}$$

where $\eta$ denotes the training rate of network learning, $\tilde{\rho}_i$ denotes the non-uniformity coefficient, and $\delta_i$ denotes the output of the hidden layer neurons.

## 4.2. Determination of parameters of RBF prediction model

The comprehensive analysis of extensive laboratory experiments and engineering surveillance of secondary steel fiber-reinforced concrete has revealed that its compressive and splitting tensile strengths are significantly influenced by three factors with notable variability: the type of steel fiber, length of the fibers, and volume of fibers. Consequently, the input layer of the enhanced RBF prediction model is designed to accommodate these three principal parameters.

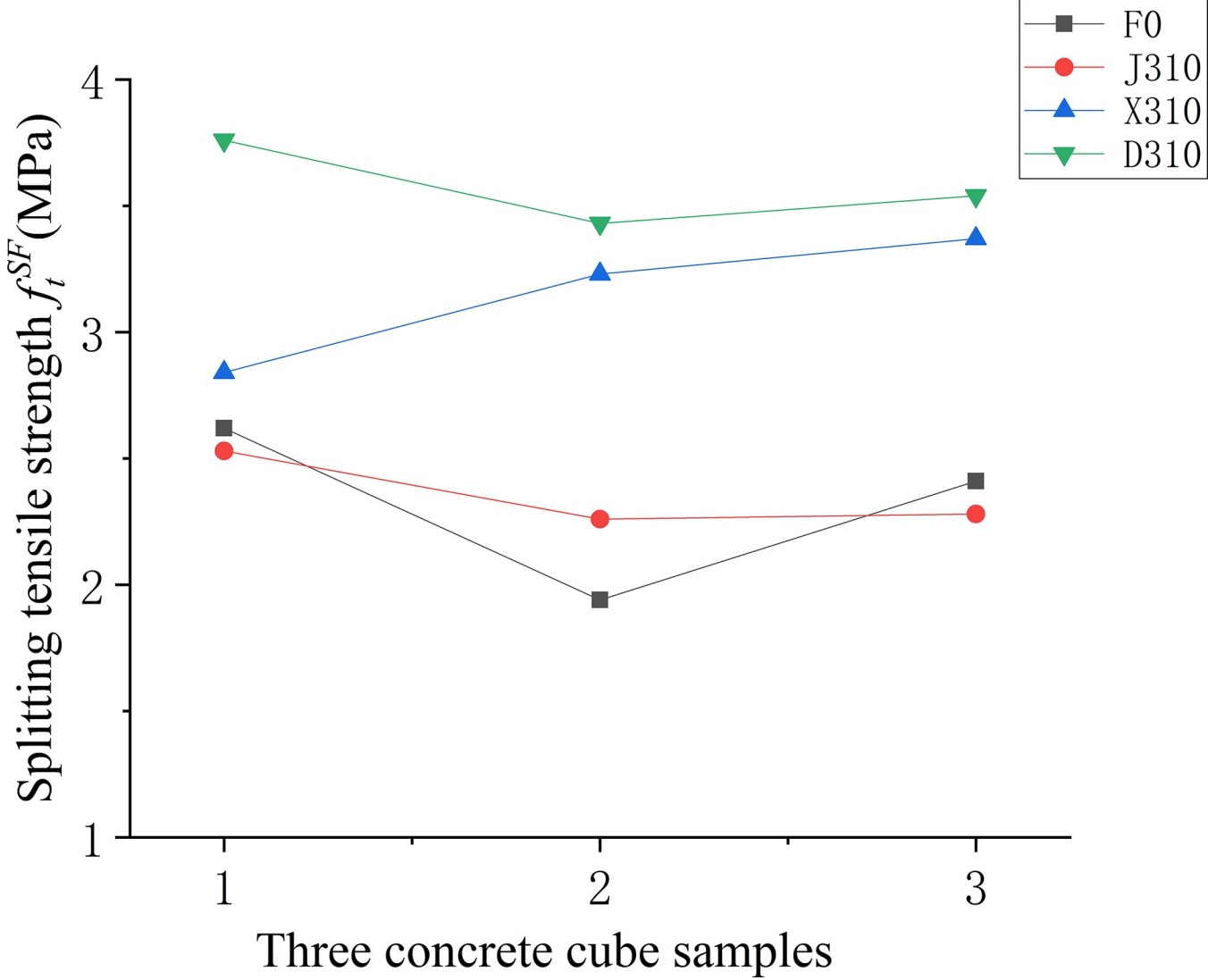

**Fig 16. Splitting tensile strength values of concrete with different fibre varieties.**

The compressive strength and splitting tensile strength constitute the foundational mechanical characteristics of concrete, serving as crucial indicators for assessing the load-bearing capacity and crack resistance of secondary steel fiber-reinforced concrete in structural engineering. In this investigation, an advanced RBF fuzzy stochastic prediction model has been developed to accurately forecast the essential mechanical properties of secondary steel fiber-reinforced concrete, considering the uncertain variability present in real-world engineering scenarios. Thus, the output layer parameters of the refined RBF prediction model are designated as the compressive and splitting tensile strengths.

According to the existence theorem of the Kolmogorov neural network mapping, the model can approximate any function when the number of hidden layer neurons is *2m+1* (*m* denotes the number of input layer parameters). Therefore, according to the actual situation of the improved RBF model in this study, it is more appropriate to set the number of hidden-layer neurons to seven [34,35].

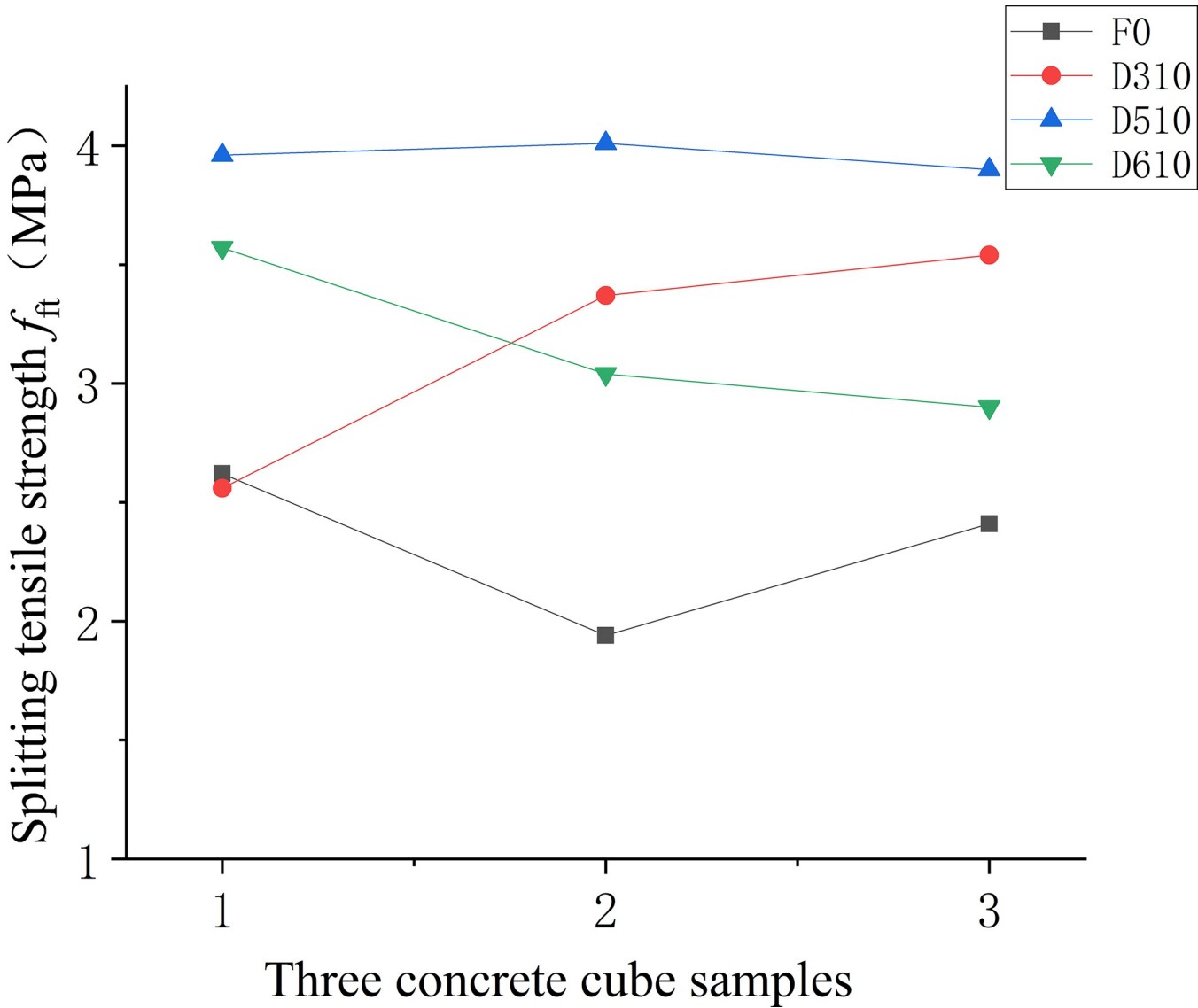

**Fig 17. Splitting tensile strength values of concrete with different fibre lengths.**

## 5. Rresults of project prediction

### 5.1. RBF fuzzy neural network engineering prediction

To accurately determine the compressive strength and splitting tensile strength of secondary steel fiber-reinforced concrete for engineering applications, this study considered a variety of steel fibers in terms of type, length, and volume used in hydraulic mass concrete projects across East China as input variables. Utilizing a trained RBF neural network model, a fuzzy random estimation was conducted for the mechanical properties of secondary steel fiber-reinforced concrete [36]. The forecast outcomes are documented in Tables 7 and 8, with corresponding visualizations of RBF prediction results displayed in Figs 20 and 21. The learning rate for the network was established at 0.2, the forgetting factor at 0.17, and the training objective for error was set below 10%. To preserve the universality of the model's input variables, steel fibers were

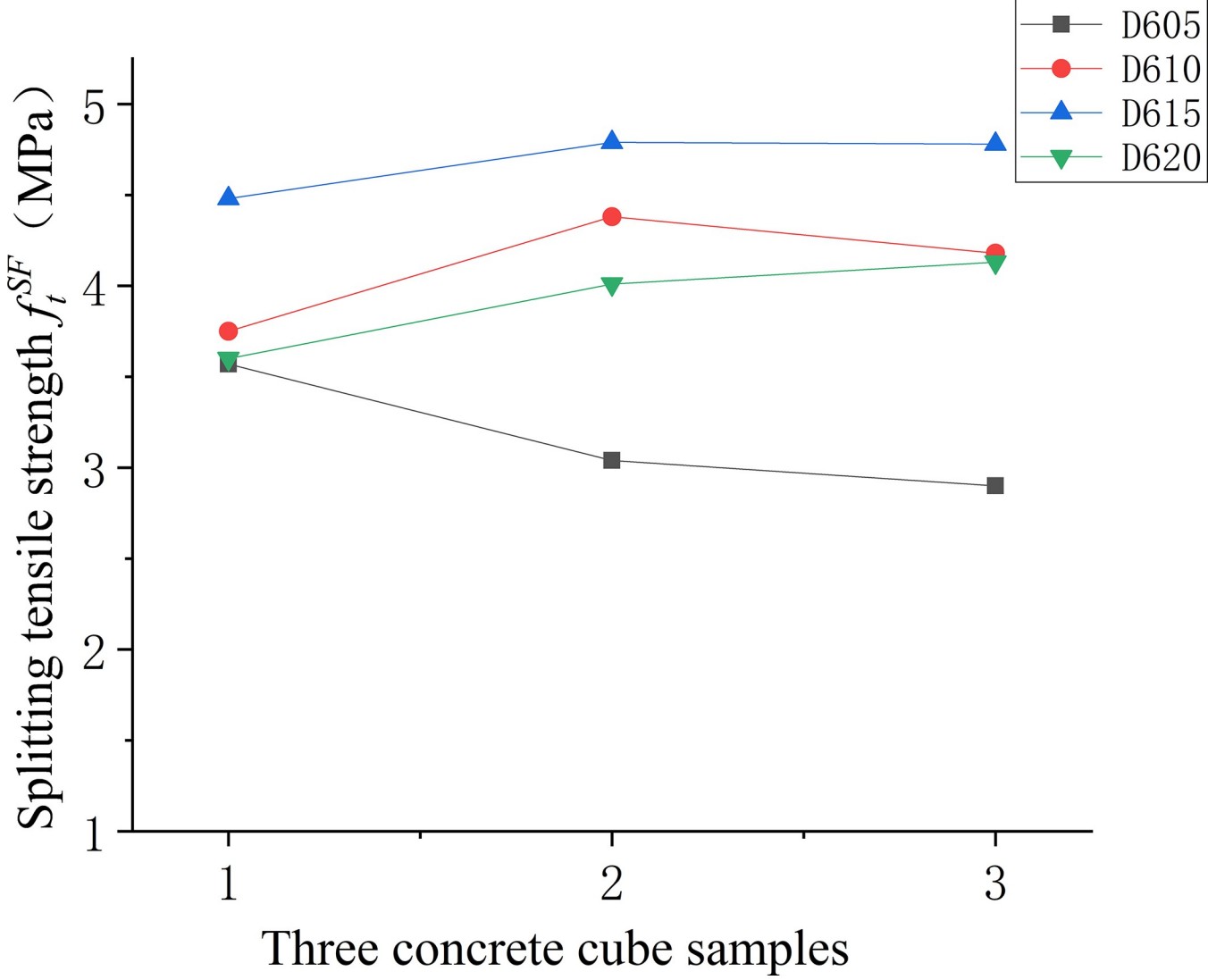

**Fig 18. Splitting tensile strength values of concrete with different fibre content.**

categorized numerically wherein the cutting type of the steel fiber was set to one, milling type was set to two, and end-hook type was set to three.

## 5.2. Analysis of prediction effect

Analysing the data from Tables 6 and 7, along with the insights from Figs 20 and 21, it is evident that the improved RBF fuzzy neural network, after training with sample data from steel fiber reinforced concrete, provides predictions for both compressive strength and splitting tensile strength that are more aligned with values measured in engineering practice than those calculated using standard formulas. The prediction error of the model is maintained below 10%. This demonstrates that the model has effectively captured the uncertain relationship between the input factors and the output properties, and thus, it can be reliably utilized to forecast the compressive strength and splitting tensile strength of secondary steel fiber-reinforced concrete in practical engineering scenarios.

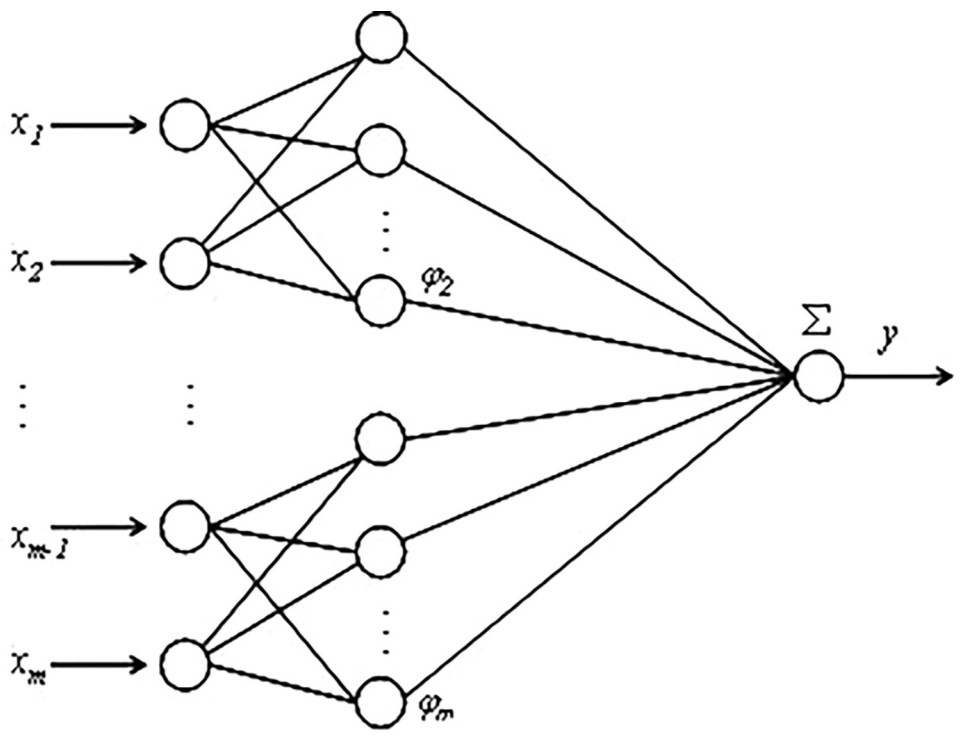

**Fig 19. RBF neural network.**

**Table 7. Results of training and learning of prediction model for compressive strength of secondary steel fiber reinforced concrete.**

| ID | Input parameter | | | Predicted value and error | | |
|---|---|---|---|---|---|---|
| | Steel fiber variety | Fiber length (m) | Fiber content (%) | Predicted compressive strength (MPa) | Compressive strength measured value (MPa) | Error (%) |
| 1 | Shear type (1) | 38 | 0.5 | 24.18 | 23.13 | 4.54 |
| 2 | Shear type (1) | 38 | 1 | 26.77 | 25.53 | 4.86 |
| 3 | Shear type (1) | 38 | 1.5 | 28.61 | 29.02 | -1.41 |
| 4 | Milling type (2) | 32 | 1 | 32.08 | 31.04 | 3.35 |
| 5 | Milling type (2) | 32 | 1.5 | 41.32 | 42.68 | -3.19 |
| 6 | Milling type (2) | 32 | 2 | 38.95 | 37.82 | 2.99 |
| 7 | End-hooked (3) | 35 | 0.5 | 30.77 | 28.75 | 7.03 |
| 8 | End-hooked (3) | 35 | 1 | 35.90 | 37.70 | -4.77 |
| 9 | End-hooked (3) | 35 | 1.5 | 38.95 | 38.08 | 2.28 |
| 10 | End-hooked (3) | 35 | 2 | 46.16 | 48.51 | -4.84 |
| 11 | End-hooked (3) | 40 | 0.5 | 31.84 | 30.67 | 3.81 |
| 12 | End-hooked (3) | 40 | 1 | 36.65 | 37.09 | -1.19 |
| 13 | End-hooked (3) | 40 | 1.25 | 39.84 | 38.56 | 3.32 |
| 14 | End-hooked (3) | 40 | 2 | 37.42 | 39.92 | -6.26 |
| 15 | End-hooked (3) | 50 | 0.5 | 35.38 | 36.07 | -1.91 |
| 16 | End-hooked (3) | 50 | 1 | 36.29 | 34.54 | 5.07 |
| 17 | End-hooked (3) | 50 | 1.5 | 44.96 | 46.09 | -2.45 |
| 18 | End-hooked (3) | 50 | 2 | 38.88 | 38.18 | 1.83 |
| 19 | End-hooked (3) | 60 | 0.5 | 36.84 | 38.86 | -5.20 |
| 20 | End-hooked (3) | 60 | 1 | 41.75 | 43.87 | -4.83 |
| 21 | End-hooked (3) | 60 | 1.25 | 41.96 | 38.61 | 8.68 |
| 22 | End-hooked (3) | 60 | 1.5 | 42.43 | 45.01 | -5.73 |

**Table 8. Results of training for prediction model of splitting tensile strength of secondary steel fiber reinforced concrete.**

| ID | Input parameter | | | Predicted value and error | | |
|----|-----------------|--|--|---------------------------|--|--|
| | Steel fiber variety | Fiber length (m) | Fiber content (%) | Predicted compressive strength (MPa) | Compressive strength measured value (MPa) | Error (%) |
| 1 | Shear type (1) | 38 | 0.5 | 2.09 | 2.22 | -5.86 |
| 2 | Shear type (1) | 38 | 1 | 2.54 | 2.6 | -2.31 |
| 3 | Shear type (1) | 38 | 1.5 | 3.12 | 2.86 | 9.09 |
| 4 | Milling type (2) | 32 | 1 | 2.55 | 2.52 | 1.19 |
| 5 | Milling type (2) | 32 | 1.5 | 3.08 | 3.28 | -6.10 |
| 6 | Milling type (2) | 32 | 2 | 3.66 | 3.98 | -8.04 |
| 7 | End-hooked (3) | 35 | 0.5 | 3.19 | 3.36 | -5.06 |
| 8 | End-hooked (3) | 35 | 1 | 3.82 | 3.78 | 1.06 |
| 9 | End-hooked (3) | 35 | 1.5 | 4.39 | 4.09 | 7.33 |
| 10 | End-hooked (3) | 35 | 2 | 5.01 | 5.38 | -6.88 |
| 11 | End-hooked (3) | 40 | 0.5 | 3.93 | 3.68 | 6.79 |
| 12 | End-hooked (3) | 40 | 1 | 4.5 | 4.64 | -3.02 |
| 13 | End-hooked (3) | 40 | 1.25 | 4.29 | 4.55 | -5.71 |
| 14 | End-hooked (3) | 40 | 2 | 4.86 | 5.13 | -5.26 |
| 15 | End-hooked (3) | 50 | 0.5 | 3.78 | 4.09 | -7.58 |
| 16 | End-hooked (3) | 50 | 1 | 4.5 | 4.2 | 7.14 |
| 17 | End-hooked (3) | 50 | 1.5 | 4.87 | 4.93 | -1.22 |
| 18 | End-hooked (3) | 50 | 2 | 5.56 | 6.07 | -8.40 |
| 19 | End-hooked (3) | 60 | 0.5 | 4.23 | 4.01 | 5.49 |
| 20 | End-hooked (3) | 60 | 1 | 4.40 | 4.04 | 8.91 |
| 21 | End-hooked (3) | 60 | 1.25 | 5.31 | 5.41 | -1.85 |
| 22 | End-hooked (3) | 60 | 1.5 | 4.89 | 5.11 | -4.31 |

## 5.3. Comparison of algorithm efficiency

The comparative analysis of the improved RBF fuzzy neural network algorithm, conventional RBF algorithm, and least squares method was conducted to predict compressive strength and splitting tensile strength, utilizing a dataset of 200 iterations. The results of this comparison are depicted in Fig 22. The experimental setup was hosted on a platform with the following specifications: an Intel Core i5-6500 CPU, 128 GB of RAM, an 800 GB hard drive, and a 1000M network card. The operating system used was Windows 10 SP3, and the computations were performed using MATLAB 2010B as the development environment.

Observations indicate that as the number of iterations increases, the improved RBF fuzzy neural network algorithm demonstrates faster convergence (increased by 15%) with smaller error margins (less than 10%). It exhibits superior robustness, efficiency, and convergence capabilities in comparison to the conventional RBF algorithm and the least squares method.

## 6. Conclusions

1. The milled steel fibers, owing to their unique shape, enhanced the compressive strength more effectively than the end-hook steel fibers. The performance of end-hook steel fibers showed an initial increase in compressive strength with the augmentation of fiber length and content. Notably, when the fiber length exceeded 50 mm and content surpassed 1%, the benefit of additional fibers diminished. It was observed that milled steel fiber enhanced compressive strength significantly more than the end-hook steel fiber. For the end-hook

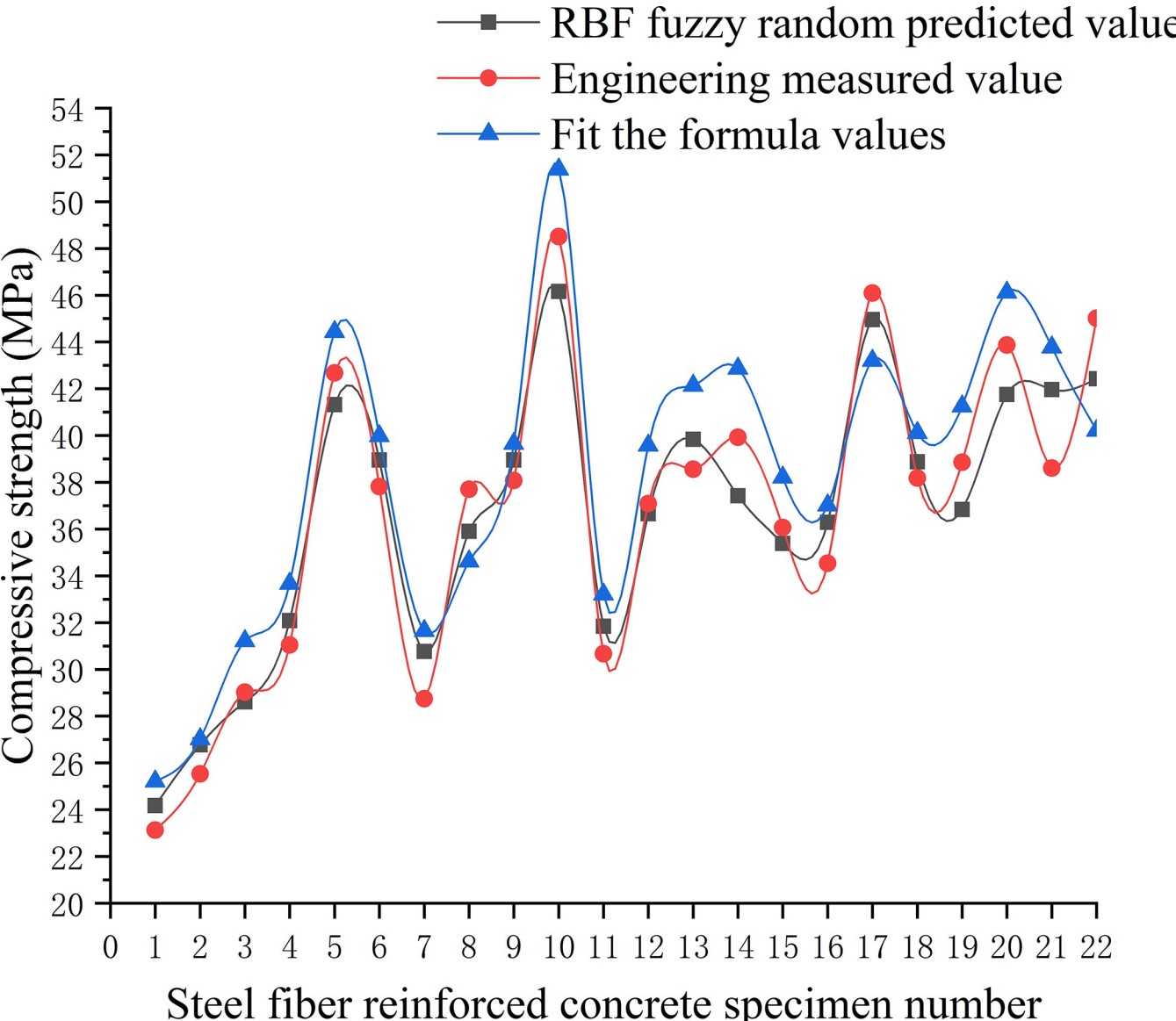

**Fig 20. Analysis of compressive strength prediction effect of RBF model.**

steel fiber, the compressive strength initially rose with increments in fiber length and content, yet this trend reversed beyond certain thresholds. Specifically, a fiber length greater than 50 mm and a content exceeding 1% resulted in diminishing returns regarding additional fiber enhancements.

2. The inclusion of steel fibers resulted in a pronounced improvement in splitting tensile strength as opposed to compressive strength. This enhancement initially increased with the addition of longer fibers and higher dosages but eventually plateaued and declined. Optimal reinforcement was achieved with fibers measuring 50 mm in length and with lf/Dmax ratio of 1.25 for the given dosage. In comparison, when the fiber content was at 1.5% and the length at 60 mm, the performance gains were less notable. In concrete of lower strength grades, the failure mode often involves the pulling out and rupture of fibers, thus the influence of fiber length and dosage on the mechanical properties is more pronounced.

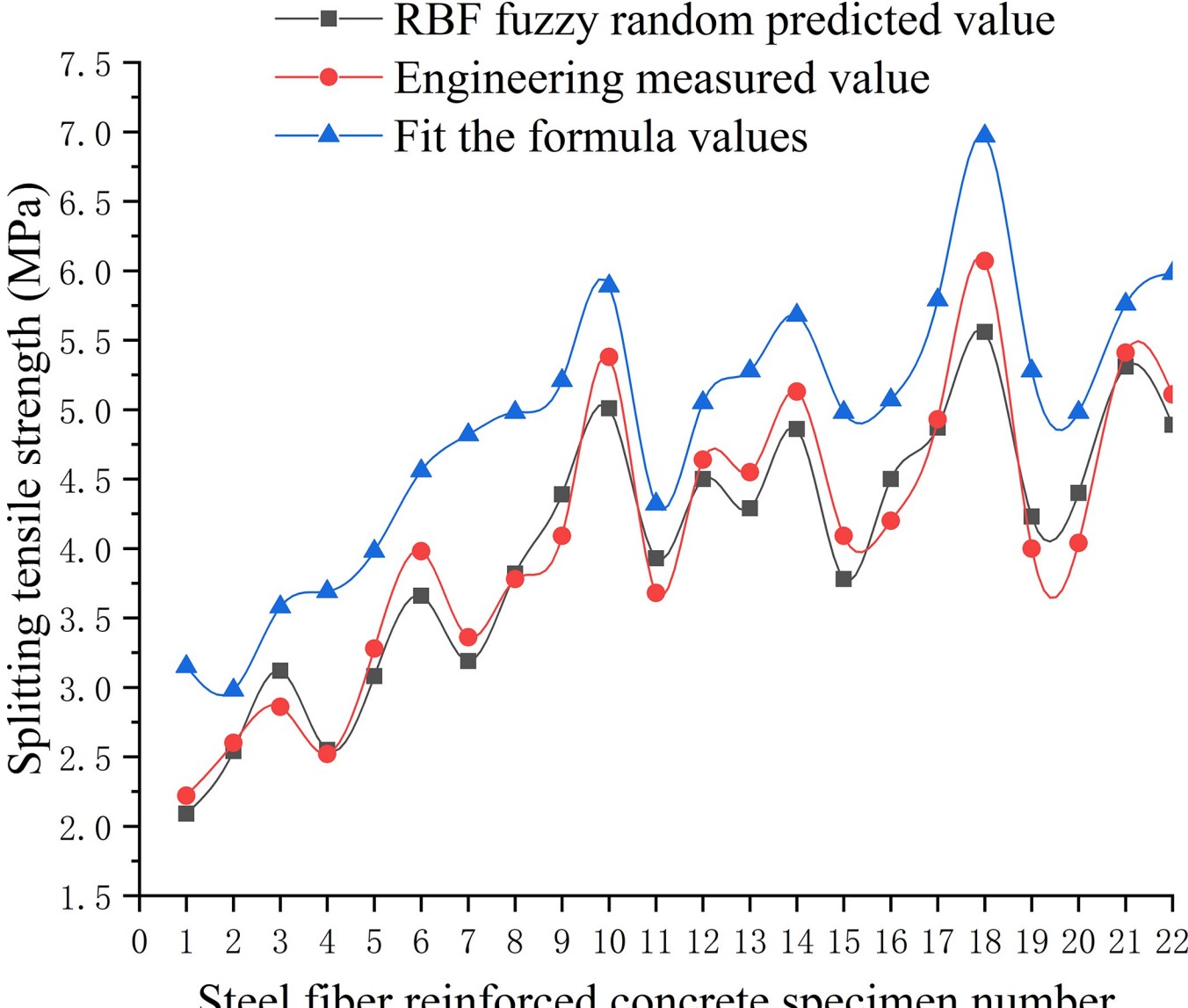

**Fig 21. Analysis of splitting tensile strength prediction effect of RBF model.**

3. Considering the variables of steel fiber type, length, and content, the compressive and splitting tensile strengths of concrete exhibit considerable variability. To address this, an enhanced RBF fuzzy neural network prediction model was developed by refining the learning strategies for central value and weight adjustments. This model uses the type, length, and content of steel fibers as input variables and outputs predicted values for compressive and splitting tensile strength. The establishment of this RBF fuzzy neural network model offers a novel approach for analysing the uncertain characteristics of the mechanical strengths of secondary steel fiber-reinforced concrete.

4. The RBF fuzzy neural network model was employed to forecast the compressive and splitting tensile strengths of steel fiber-reinforced concrete. The predictive convergence rate of the model is increased by 15%, accuracy was corroborated by a less than 10% deviation between forecasted and actual measurements, which surpasses the precision of

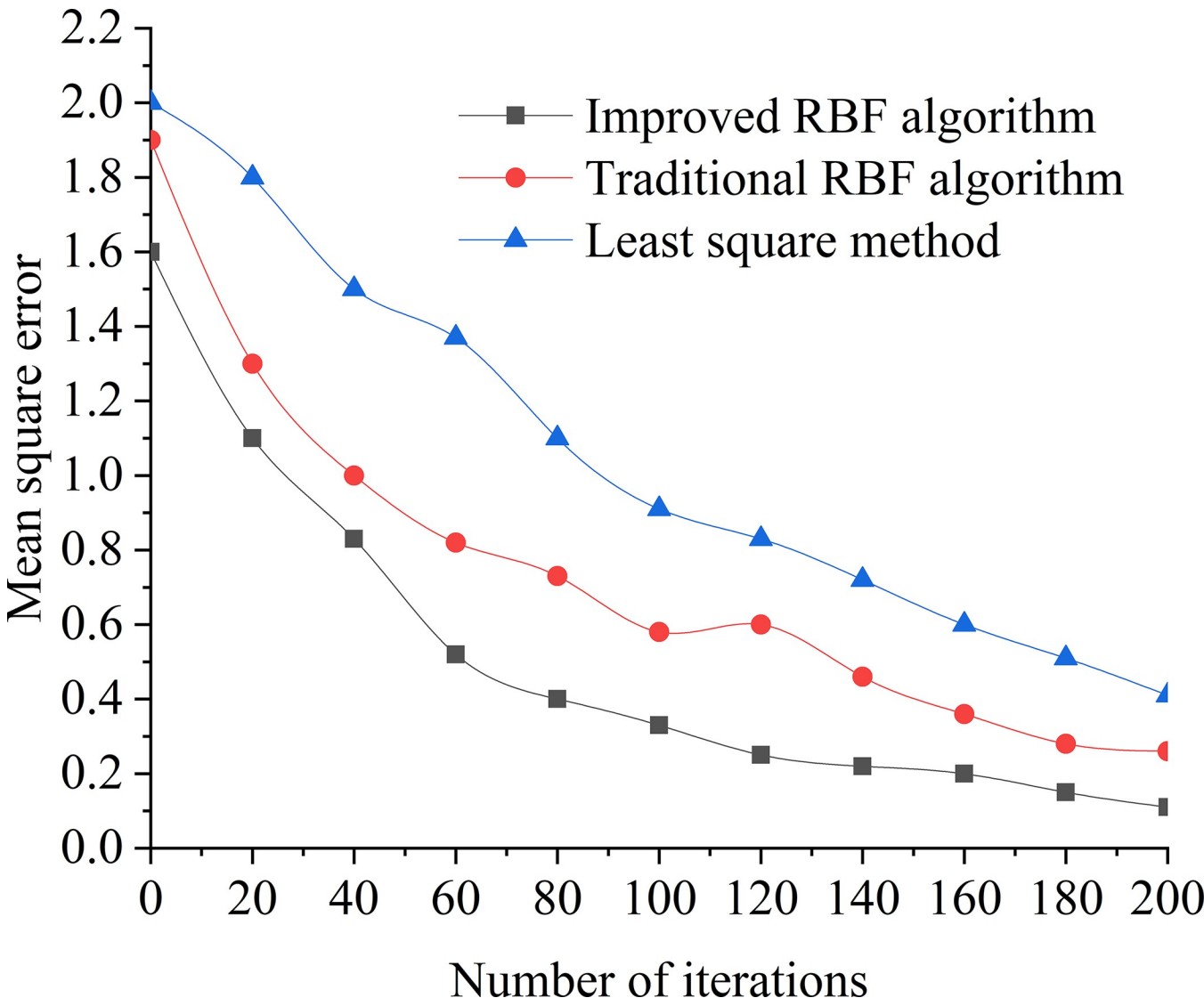

**Fig 22. Comparison chart of algorithm efficiency.**

conventional formulas. Moreover, the refined algorithm of the model demonstrates high efficiency and accuracy, offering reliable technical backing for the secure implementation of substantial steel fiber concrete ventures, including rock anchor beams and similar constructions.

5. Since the test data of secondary steel fiber reinforced concrete is not comprehensive enough, the RBF fuzzy neural network model has not fully utilized the advantages of prediction. Next, the model will be further optimized by supplementing test data under different engineering conditions and citing literature data under similar conditions.

## Supporting information

**S1 Dataset.**

(XLS)

## Acknowledgments

We are grateful to the volunteers who participated in the Ginkgo Evaluation of Memory Study.

## Author Contributions

**Conceptualization:** Song Ling.

**Data curation:** Li Yongheng.

**Funding acquisition:** Yao Yafeng.

**Methodology:** Du Chengbin, Yao Yafeng.

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
