## [Decision Letter · Decision Letter 0]

6 Dec 2023

PONE-D-23-39113Analysis and prediction of compressive and split-tensile strength of secondary steel fibre reinforced concrete based on RBF fuzzy neural network modelPLOS ONE

Dear Dr. Lin,

Thank you for submitting your manuscript to PLOS ONE. After careful consideration, we feel that it has merit but does not fully meet PLOS ONE’s publication criteria as it currently stands. Therefore, we invite you to submit a revised version of the manuscript that addresses the points raised during the review process.

We look forward to receiving your revised manuscript.

Kind regards,

Paul Awoyera

Academic Editor

PLOS ONE

Journal Requirements:

   "Funding: This work was supported by the National Natural Science Foundation of China [Grant Numbers 51874005, 51474004,YYF]; Nantong Municipal Science and Technology Program of China [grant number JCZ2022088,SL]; and the Research project of Nantong Vocational University of China[Grant Numbers 22ZK01,YYF]."

7. PLOS requires an ORCID iD for the corresponding author in Editorial Manager on papers submitted after December 6th, 2016. Please ensure that you have an ORCID iD and that it is validated in Editorial Manager. To do this, go to ‘Update my Information’ (in the upper left-hand corner of the main menu), and click on the Fetch/Validate link next to the ORCID field. This will take you to the ORCID site and allow you to create a new iD or authenticate a pre-existing iD in Editorial Manager. Please see the following video for instructions on linking an ORCID iD to your Editorial Manager account: https://www.youtube.com/watch?v=_xcclfuvtxQ

8. Please ensure that you refer to Figure 4 in your text as, if accepted, production will need this reference to link the reader to the figure.

Reviewers' comments:

Reviewer's Responses to Questions

**Comments to the Author**

1. Is the manuscript technically sound, and do the data support the conclusions?

Reviewer #1: No

Reviewer #2: Yes

Reviewer #3: Yes

2. Has the statistical analysis been performed appropriately and rigorously? 

Reviewer #1: No

Reviewer #2: Yes

Reviewer #3: No

3. Have the authors made all data underlying the findings in their manuscript fully available?

Reviewer #1: No

Reviewer #2: Yes

Reviewer #3: No

4. Is the manuscript presented in an intelligible fashion and written in standard English?

Reviewer #1: No

Reviewer #2: Yes

Reviewer #3: Yes

5. Review Comments to the Author

Reviewer #1: The paper titled 'Analysis and prediction of compressive and split-tensile strength of secondary steel

fibre reinforced concrete based on RBF fuzzy neural network model' show no novelty for the following reseaons:

the data used to generate the models are very limited only 22 points.

the paper is written like a report and there is no criticalty.

the paper seem to have expermental work it is unclear how that was conducted and why

it is unclear why the investigation focused on spliting and compressive testing

the title is not reflective of the paper

the conclusion is poorly written

the intoduction lacks in depth review

Reviewer #2: Accurate analysis of the strength of steel-fibre-reinforced concrete (SFRC) is important for ensuring construction quality and safety. The cubic compression and splitting tensile tests of steel fibers of different varieties, lengths and contents were carried out in the manuscript. On this basis, the RBF fuzzy neural network prediction model of the strength of secondary steel fiber reinforced concrete was established. The improved prediction model is more efficient and reasonable in accuracy than the previous algorithm. Generally speaking, the structure, goals and results are clear and reasonable. Therefore, I suggest the manuscript be accepted for publication after minor revisions. Some of the specific review comments are listed.

(1) It is recommended to further refine and simplify the abstract.

(2)In order to illustrate the process of RBF network, please mark input layer, hidden layer and output layer clearly in Fig.10.

(3)The name of horizontal coordinate in Fig. 8 is not clearly described, please modify it. And verify the coordinate names in the other diagrams.

(4)The RBF fuzzy neural network was established in the manuscript to predict compressive and cleavage stresses. Among the three inputs of the improved model, how does the author consider the values of steel fiber types?

(5)The three types of steel fiber photos are not clear, so it is recommended to give the photo of a single fiber and the corresponding geometric diagram.

(6)Steel fiber can comprehensively improve the physical and mechanical properties and durability of concrete. In general, the benchmark strength is higher, it is suggested why only C25 is used in this study.

(7)The slump of concrete is small, should not be able to meet the pumping conditions. If according to the pumping concrete design, what will happen to the performance of the concrete?

(8)The writing format of strength symbol is not standard, such as fcf should write fc, SF (SF can also be placed at the superscript site).

(9)All variables should be written in italics, and the full text should be consistent.

(10)There are some mistakes on spelling, such as Hohai and Hehai.

Reviewer #3: Overall Evaluation

This paper presents an approach to predicting the strength of secondary steel fiber-reinforced concrete using an RBF fuzzy neural network model. The methodology shows promise in addressing the complexity and variability inherent in concrete strength analysis. However, the paper would benefit from a more thorough clarification of its theoretical underpinnings and a more detailed explanation of the model's application in real-world scenarios to enhance its practical relevance and applicability in the field of concrete technology.

1-Language & Structure

The paper's structure and language are mostly clear, but certain improvements could enhance its overall quality. A notable issue is the inconsistent use of British and American English spellings, such as "fibre” and "fiber." Standardizing the spelling throughout the paper would contribute to its professional presentation. A thorough proofreading is recommended to address these issues and refine the paper's language. Additionally, while the “Materials and Methods” section (Sections 2 & 4) is well-detailed, ensuring that it is distinctly separated from the results and discussion sections would improve clarity and flow.

2-Abstract

The abstract effectively summarizes the study's primary objectives and methods, focusing on the analysis of steel fiber-reinforced concrete and the use of an RBF fuzzy neural network for prediction. However, it lacks a direct statement of the research's novelty or unique contribution to the field, which is essential for highlighting the significance and relevance of the study. Additionally, the abstract could benefit from a more explicit mention of the key findings or implications of the research, providing readers with a clearer understanding of the study's impact on practical engineering applications.

3-Introduction

The introduction section of the paper presents a comprehensive and technically detailed context for the study of secondary steel fiber reinforced concrete, particularly emphasizing its application in mass hydraulic concrete and the challenges associated with crack control. The literature review is thorough, capturing the essence of current research trends and regulatory standards. However, the section would benefit from a clearer delineation of the study's unique contributions and a more gradual integration of specific technical details. The early introduction of the enhanced RBF neural network model, while innovative, calls for a more detailed justification, particularly in contrast to existing models.

•Is there a specific reason for the immediate introduction of the RBF neural network model in the introduction? A more gradual approach might be more effective.

•Would a more critical evaluation of the cited studies, particularly how they directly inform the current research’s methodology and objectives, enhance the introduction's depth?

While the current study provides an insightful analysis of steel fiber reinforced concrete, exploring the impact of various fiber types on mechanical properties, it may be beneficial to also consider recent advancements in concrete strengthening techniques. For instance, the work by Khaleel Ibrahim & Movahedi Rad (2023) on the optimal plastic behavior of RC beams strengthened by carbon fiber polymers offers valuable insights into reliability-based design approaches (http://dx.doi.org/10.3390/polym15030569). Such perspectives could potentially enrich the discussion on alternative strengthening methods and materials in future studies. In addition, the current paper's focus on predicting concrete strength using the RBF fuzzy neural network model is commendable. However, it could be interesting to explore comparisons with other advanced predictive algorithms. For instance, the study by Ghodousian et al. (2021) utilizes a modified PSO algorithm for predicting the compressive strength of pozzolanic concrete. Another research, the investigation by Oveys Ghodousian et al. (2023), which employs a fuzzy model for predicting the outcomes of shear-splitting tests in facade stone bonding, aligns well with the fuzzy approach used in the current study (https://doi.org/10.3390/buildings13051229). Incorporating a comparative analysis with such algorithms might offer a broader perspective on the efficacy of different predictive methods in concrete technology.

•How does the study address the apparent scarcity of consistent data on the compressive and splitting tensile strengths of secondary steel fiber reinforced concrete?

•Is it possible to include a preliminary discussion on why traditional models may be insufficient compared to the RBF neural network approach?

•Recommendation: Consider reorganizing the introduction to provide a more seamless transition from general context to specific research focus, enhancing coherence and readability.

•Recommendation: A clearer statement of the research objectives in the context of the identified gaps would set a more focused direction for the study.

4-Test Overview (Materials & Methods)

This section provides a foundational overview of the experimental setup for evaluating the properties of secondary steel fiber reinforced concrete. While the description of materials and the general preparation process is comprehensive, the section falls short in detailing the specific mix designs and fiber ratios for each test specimen. This lack of specificity may hinder the ability to fully understand the experimental conditions and to replicate the study. Enhancing the clarity and detail of the mix designs and incorporating more explicit procedural steps would substantially improve the section's usefulness and scientific rigor.

•Why are specific mix design IDs and corresponding fiber ratios for each specimen not included? How does this omission affect the clarity and replicability of the research?

•Could the authors provide a more detailed explanation of the decision-making process behind the chosen mix ratios and fiber types, including their impact on concrete properties?

•Is there an explanation for the observed fiber clumping at higher content levels, and how was this issue addressed during the experiment?

•How do the chosen specimen preparation and testing methods align with the study's objectives, and are these methods adequately detailed for replication purposes?

•Recommendation: Consider adding a detailed breakdown of each mix design with specific IDs and fiber ratios to enhance clarity and enable replication of the study.

•Recommendation: Expand the procedural steps in specimen preparation and testing, including mixing times, curing conditions, and specific testing protocols, to provide a comprehensive understanding of the experimental setup.

5-Conclusion

The conclusion effectively summarizes the key findings of the study, highlighting the influence of different steel fiber types on the mechanical properties of concrete and the development of an enhanced RBF fuzzy neural network model. However, it would benefit from streamlining to eliminate repetitive content, particularly in points (1) and (2). Additionally, incorporating a discussion about the limitations of the current study and suggesting potential directions for future research would provide a more comprehensive and balanced view.

6. PLOS authors have the option to publish the peer review history of their article (what does this mean?). If published, this will include your full peer review and any attached files.

Reviewer #1: No

Reviewer #2: No

Reviewer #3: No

---

## [Author Response · Author response to Decision Letter 0]

22 Jan 2024

Response to the Referees’ Comments 

Manuscript Number: PONE-D-23-39113

Title: Analysis and prediction of compressive and split-tensile strength of secondary steel fiber reinforced concrete based on RBF fuzzy neural network model 

Journal: PLoS One

Author(s): Song Ling, Du Chengbin, Yao Yafeng and LI Yongheng 

Dear Prof. Paul Awoyera, Editors and Reviewers:

Thank you for your letter and for the reviewers’comments concerning our manuscript entitled “Analysis and prediction of compressive and split-tensile strength of secondary steel fibre reinforced concrete based on RBF fuzzy neural network model”(Number: PONE-D-23-39113). These comments are all valuable and very helpful for revising and improving our manuscript, as well as the important guiding significance to our researches. We have studied comments carefully and have made corrections which we hope meet with approval. Revised portion are highlighted in red in the revised manuscript. The main corrections in the revised manuscript and the responds to the reviewer’s comments are as following:

Reviewer 2:.

Reviewer’s comments:

Accurate analysis of the strength of steel-fibre-reinforced concrete (SFRC) is important for ensuring construction quality and safety. The cubic compression and splitting tensile tests of steel fibers of different varieties, lengths and contents were carried out in the manuscript. On this basis, the RBF fuzzy neural network prediction model of the strength of secondary steel fiber reinforced concrete was established. The improved prediction model is more efficient and reasonable in accuracy than the previous algorithm. Generally speaking, the structure, goals and results are clear and reasonable. Therefore, I suggest the manuscript be accepted for publication after minor revisions. Some of the specific review comments are listed.

(1)It is recommended to further refine and simplify the abstract.

Authors’ response:

Thanks for the reviewers' valuable suggestions. The abstract have been simplified and refined according to the main content, and the revised abstract is as follows:

Accurate analysis of the strength of steel-fiber-reinforced concrete (SFRC) is important for ensuring construction quality and safety. Cube compression and splitting tensile tests of steel fiber with different varieties, lengths, and dosages were performed, and the effects of different varieties, lengths, and dosages on the compressive and splitting properties of secondary concrete were obtained. It was determined that the compression and splitting strengths of concrete could be effectively improved by the addition of end-hooked and milled steel fibers. The compressive and splitting strengths of concrete can be enhanced by increasing the fiber length and content. However, concrete also exhibits obvious uncertainty owing to the comprehensive influence of steel fiber variety, fiber length, and fiber content. In order to solve this engineering uncertainty, the traditional RBF neural network is improved by using central value and weight learning strategy especially. On this basis, the RBF fuzzy neural network prediction model of the strength of secondary steel fiber-reinforced concrete was innovatively established with the type, length and content of steel fiber as input information and the compressive strength and splitting tensile strength as output information. In order to further verify the engineering reliability of the prediction model, the compressive strength and splitting tensile strength of steel fiber reinforced concrete with rock anchor beams are predicted by the prediction model. The results show that the convergence rate of the prediction model is increased by 15%, and the error between the predicted value and the measured value is less than 10%, which is more efficient and accurate than the traditional one. Additionally, the improved model algorithm is efficient and reasonable, providing technical support for the safe construction of large-volume steel fiber concrete projects, such as rock anchor beams. The fuzzy random method can also be applied to similar engineering fields.

(2)In order to illustrate the process of RBF network, please mark input layer, hidden layer and output layer clearly in Fig.10.

Authors’ response:

Fig.10 in the manuscript has already marked the input layer, hidden layer and output layer in detail according to the content.

Input layer Hidden layer Output layer

Figure 10. RBF neural network

(3)The name of horizontal coordinate in Fig. 8 is not clearly described, please modify it. And verify the coordinate names in the other diagrams.

Authors’ response:

According to the content of compressive tests of different varieties of fiber concrete cubes in the manuscript, the horizontal coordinate in Fig. 8 is changed to "concrete cube specimen number". The corresponding corrections have been made in Fig. 9, as detailed in the revised manuscript.

(4)The RBF fuzzy neural network was established in the manuscript to predict compressive and cleavage stresses. Among the three inputs of the improved model, how does the author consider the values of steel fiber types?

Authors’ response:

According to the engineering application of mass fiber reinforced concrete, the fuzzy prediction of compressive and splitting tensile strength is mainly carried out for Shear type, milling type and End-hooked fiber reinforced concrete. Considering the three different fiber types, the input values have been numeralized to 1,2, and 3 respectively, and the results do not affect the final predicted performance.

(5)The three types of steel fiber photos are not clear, so it is recommended to give the photo of a single fiber and the corresponding geometric diagram.

Authors’ response:

According to reviewers' advice, Figure 2 is replaced with physical drawings and geometric dimensions graph of three different steel fibers, as shown in the figure below.

（a）Physical drawing （b）Geometric graph

Figure 1. Shapes of three different types of steel fibers

(6)Steel fiber can comprehensively improve the physical and mechanical properties and durability of concrete. In general, the benchmark strength is higher, it is suggested why only C25 is used in this study.

Authors’ response:

The main purpose of this paper is to analyze and predict the performance of large volume fiber reinforced concrete in hydraulic engineering. Through engineering investigation, it is found that low strength concrete is commonly used in these projects. Therefore, this manuscript takes C25 as an example to study.

(7)The slump of concrete is small, should not be able to meet the pumping conditions. If according to the pumping concrete design, what will happen to the performance of the concrete?

Authors’ response:

This manuscript mainly studies the properties of steel fiber reinforced concrete under non-pumping construction. If the pumped concrete design is followed, it is necessary to reduce the corresponding fiber parameters and increase the work ability of fiber concrete, so as to meet the construction requirements of pumped concrete (slump 8-18cm). This problem will be further studied by the research group.

(8)The writing format of strength symbol is not standard, such as fcf should write fc, SF (SF can also be placed at the superscript site).

Authors’ response:

Thanks for the reviewers' valuable suggestions. The authors carefully checked and standardized the writing format of strength symbol in the whole manuscript. The partially modified versions are shown in Figure 8 and Table 6.

（a）Different fiber varieties （b）Different fiber lengths （c）Different fiber content

Figure 8. Compressive strength values of a group of three specimens of fiber reinforced concrete

Table 6. Test results of splitting tensile strength of specimens

Different working conditions Sample 1

ftSF（MPa） Sample 2

ftSF（MPa） Sample 3

ftSF（MPa）

F0 2.62 1.94 2.41 

J310 2.53 2.26 2.28 

X310 2.84 3.23 3.37 

D310 3.76 3.43 3.54 

D510 4.67 4.57 4.47 

D605 3.57 3.04 2.90 

D610 3.75 4.38 4.18 

D615 4.48 4.79 4.78 

D620 3.60 4.01 4.13 

(9)All variables should be written in italics, and the full text should be consistent.

Authors’ response:

Thanks for the reviewers' valuable suggestions. All variables have been written in italics. In addition, the authors conducted a consistency check on the variables in the manuscript, as shown in the revised manuscript.

(10)There are some mistakes on spelling, such as Hohai and Hehai.

Authors’ response:

I'm sorry for making some spelling mistakes. In the revised manuscript, the spelling has been standardized and checked by the author. In addition, the manuscript has been fully polished by native speakers, as shown in the revised manuscript.

Reviewer 3:

Reviewer’s comments:

Reviewer #3:Overall Evaluation

This paper presents an approach to predicting the strength of secondary steel fiber-reinforced concrete using an RBF fuzzy neural network model. The methodology shows promise in addressing the complexity and variability inherent in concrete strength analysis. However, the paper would benefit from a more thorough clarification of its theoretical underpinnings and a more detailed explanation of the model's application in real-world scenarios to enhance its practical relevance and applicability in the field of concrete technology.

1-Language & Structure

The paper's structure and language are mostly clear, but certain improvements could enhance its overall quality. A notable issue is the inconsistent use of British and American English spellings, such as "fibre” and "fiber." Standardizing the spelling throughout the paper would contribute to its professional presentation. A thorough proofreading is recommended to address these issues and refine the paper's language. Additionally, while the “Materials and Methods” section (Sections 2 & 4) is well-detailed, ensuring that it is distinctly separated from the results and discussion sections would improve clarity and flow.

Authors’ response:

In order to ensure the consistency of English expression throughout the manuscript, "fibre "in the original manuscript is replaced with "fiber."

In order to improve the clarity of the paper structure, the title of Section 2 is changed to "materials", and the title of Section 3 is changed to" Test methods and analysis ". Section5 should be titled "Rresults of project prediction". The content of the corresponding section has also been adjusted, as detailed in the revised manuscript.

Based on reviewer’s advice, the English language and the text explanation of the manuscript had been made a major revision. In addition, the manuscript had been fully polished by a native language person, as shown in the revised manuscript.

2-Abstract

The abstract effectively summarizes the study's primary objectives and methods, focusing on the analysis of steel fiber-reinforced concrete and the use of an RBF fuzzy neural network for prediction. However, it lacks a direct statement of the research's novelty or unique contribution to the field, which is essential for highlighting the significance and relevance of the study. Additionally, the abstract could benefit from a more explicit mention of the key findings or implications of the research, providing readers with a clearer understanding of the study's impact on practical engineering applications.

Authors’ response:

The authors have reorganized and refined the abstracts according to the suggestions of the reviewers. Focusing on the core content of the research, the key findings and unique contributions to the field are highlighted, and the revised abstract is as follows:

Accurate analysis of the strength of steel-fiber-reinforced concrete (SFRC) is important for ensuring construction quality and safety. Cube compression and splitting tensile tests of steel fiber with different varieties, lengths, and dosages were performed, and the effects of different varieties, lengths, and dosages on the compressive and splitting properties of secondary concrete were obtained. It was determined that the compression and splitting strengths of concrete could be effectively improved by the addition of end-hooked and milled steel fibers. The compressive and splitting strengths of concrete can be enhanced by increasing the fiber length and content. However, concrete also exhibits obvious uncertainty owing to the comprehensive influence of steel fiber variety, fiber length, and fiber content. In order to solve this engineering uncertainty, the traditional RBF neural network is improved by using central value and weight learning strategy especially. On this basis, the RBF fuzzy neural network prediction model of the strength of secondary steel fiber-reinforced concrete was innovatively established with the type, length and content of steel fiber as input information and the compressive strength and splitting tensile strength as output information. In order to further verify the engineering reliability of the prediction model, the compressive strength and splitting tensile strength of steel fiber reinforced concrete with rock anchor beams are predicted by the prediction model. The results show that the convergence rate of the prediction model is increased by 15%, and the error between the predicted value and the measured value is less than 10%, which is more efficient and accurate than the traditional one. Additionally, the improved model algorithm is efficient and reasonable, providing technical support for the safe construction of large-volume steel fiber concrete projects, such as rock anchor beams. The fuzzy random method can also be applied to similar engineering fields.

3-Introduction

The introduction section of the paper presents a comprehensive and technically detailed context for the study of secondary steel fiber reinforced concrete, particularly emphasizing its application in mass hydraulic concrete and the challenges associated with crack control. The literature review is thorough, capturing the essence of current research trends and regulatory standards. However, the section would benefit from a clearer delineation of the study's unique contributions and a more gradual integration of specific technical details. The early introduction of the enhanced RBF neural network model, while innovative, calls for a more detailed justification, particularly in contrast to existing models.

•Is there a specific reason for the immediate introduction of the RBF neural network model in the introduction? A more gradual approach might be more effective.

•Would a more critical evaluation of the cited studies, particularly how they directly inform the current research’s methodology and objectives, enhance the introduction's depth?

While the current study provides an insightful analysis of steel fiber reinforced concrete, exploring the impact of various fiber types on mechanical properties, it may be beneficial to also consider recent advancements in concrete strengthening techniques. For instance, the work by Khaleel Ibrahim & Movahedi Rad (2023) on the optimal plastic behavior of RC beams strengthened by carbon fiber polymers offers valuable insights into reliability-based design approaches (http://dx.doi.org/10.3390/polym15030569). Such perspectives could potentially enrich the discussion on alternative strengthening methods and materials in future studies. In addition, the current paper's focus on predicting concrete strength using the RBF fuzzy neural network model is commendable. However, it could be interesting to explore comparisons with other advanced predictive algorithms. For instance, the study by Ghodousian et al. (2021) utilizes a modified PSO algorithm for predicting the compressive strength of pozzolanic concrete. Another research, the investigation by Oveys Ghodousian et al. (2023), which employs a fuzzy model for predicting the outcomes of shear-splitting tests in facade stone bonding, aligns well with the fuzzy approach used in the current study (https://doi.org/10.3390/buildings13051229). Incorporating a comparative analysis with such algorithms might offer a broader perspective on the efficacy of different predictive methods in concrete technology.

•How does the study address the apparent scarcity of consistent data on the compressive and splitting tensile strengths of secondary steel fiber reinforced concrete?

•Is it possible to include a preliminary discussion on why traditional models may be insufficient compared to the RBF neural network approach?

•Recommendation: Consider reorganizing the introduction to provide a more seamless transition from general context to specific research focus, enhancing coherence and readability.

•Recommendation: A clearer statement of the research objectives in the context of the identified gaps would set a more focused direction for the study.

Authors’ response:

Thanks for the reviewers' valuable suggestions. According to the advice, the authors have carefully revised the Introduction part.

1.A special paragraph is added to introduce the application of artificial intelligence algorithms in concrete engineering. This paragraph summarizes and analyzes the characteristics of current algorithm application, especially the shortcomings in the face of uncertain engineering prediction. The advantage of fuzzy RFB neural network algorithm is introduced in this paper, which makes the transition more reasonable.The specific contents are as follows:

In view of the uncertainty distribution of steel fiber reinforced concrete strength in practical engineering, some scholars try to use intelligent algorithms to make comprehensive analysis in order to improve efficiency. Oveys et al. [26] presents an investigation into the bond strength of travertine, granite, and marble, to a concrete substrate using a shear-splitting test. Based on the findings, a novel fuzzy logic approach was proposed to predict the bond strength. Wang et al. [27] established random forest (RF) to predict UCS by analyzing and comparing five traditional models: RF, multiple regression analysis (MR), backpropagation neural network (BPNN), extreme learning Machine (ELM) and support vector regression (SVR). Pouria et al. [28] compared traditional backpropagation algorithms (LM), differential evolution (DE), and particle swarm optimization (PSO). On this basis, artificial neural network (ANN) technology is combined with a robust optimization technique PSOTD to predict the CS of RHA concrete. Through the analysis of these documents, it is found that most of the current intelligent algorithm models of engineering prediction only focus on randomness or fuzziness of engineering, and do not consider the two comprehensively. This will cause results to deviate from reality. Therefore, in view of the influence of different types, lengths and quantities of steel fibers on the performance of secondary steel fiber reinforced concrete, the traditional RBF neural network is improved, and the optimized fuzzy RBF neural network is established to be a more effective tool for the performance prediction of steel fiber reinforced concrete.

2.In order to deeply analyze the mechanical properties of steel fiber reinforced concrete and explore the influence of various fiber types on the mechanical properties, the latest progress of concrete reinforcement technology is added in the introduction section. For example, which Dr. Khaleel Ibrahim published in 2023, about the best plastic behavior of carbon fiber polymer reinforced RC beam research (http://dx.doi.org/10.3390/polym15030569).

3. In order to effectively analyze the application of artificial intelligence algorithms to engineering uncertain problems, the introduction also analyzes and compares other advanced prediction algorithms. For example, Ghodousian et al. used improved particle swarm optimization to predict the compressive strength of pozzolanic concrete. Oveys Ghodousian by establishing a fuzzy model to predict the facade stone adhesive shear fracturing in test results (https://doi.org/10.3390/buildings13051229). This section, combined with these algorithms through comparative analysis, may provide a broader perspective on the effectiveness of different prediction methods in concrete technology. The corresponding reference section also adds to these research results.

4. At present, the test data of secondary steel fiber reinforced concrete is few, which is the reason why this paper intends to use the improved intelligent algorithm to predict the performance. The improved RBF neural network can reasonably reflect the compressive strength and splitting strength in accordance with the actual working conditions under the condition that the basic experimental data are few. In the next step, the authors will further optimize the model by supplementing the test data of different engineering conditions and citing the documents data of similar conditions.

5.According to reviewers' suggestion, the Introduction section has been reorganized. The revised section provides a more seamless transition from general context to specific research focus, enhancing coherence and readability. The last paragraph of this section more clearly states the research objectives of this paper and sets a more focused direction for the research.

4-Test Overview (Materials & Methods)

This section provides a foundational overview of the experimental setup for evaluating the properties of secondary steel fiber reinforced concrete. While the description of materials and the general preparation process is comprehensive, the section falls short in detailing the specific mix designs and fiber ratios for each test specimen. This lack of specificity may hinder the ability to fully understand the experimental conditions and to replicate the study. Enhancing the clarity and detail of the mix designs and incorporating more explicit procedural steps would substantially improve the section's usefulness and scientific rigor.

•Why are specific mix design IDs and corresponding fiber ratios for each specimen not included? How does this omission affect the clarity and replicability of the research?

•Could the authors provide a more detailed explanation of the decision-making process behind the chosen mix ratios and fiber types, including their impact on concrete properties?

•Is there an explanation for the observed fiber clumping at higher content levels, and how was this issue addressed during the experiment?

•How do the chosen specimen preparation and testing methods align with the study's objectives, and are these methods adequately detailed for replication purposes?

•Recommendation: Consider adding a detailed breakdown of each mix design with specific IDs and fiber ratios to enhance clarity and enable replication of the study.

•Recommendation: Expand the procedural steps in specimen preparation and testing, including mixing times, curing conditions, and specific testing protocols, to provide a comprehensive understanding of the experimental setup.

Authors’ response:

Thanks for the reviewers' valuable suggestions. In order to improve the clarity and detail of the mix design and to include more explicit procedural steps, the Materials & Methods sections are modified as follows:

1.Table 3 (Mix ratio of secondary steel fiber reinforced concrete) is added to this section, which includes the id of each sample, the corresponding Fiber ratio, Sand ratio, and the Mix ratio of Water, Cement and Sand.

2.This study comes from the real project cases of enterprises, and this mix ratio is commonly used in local practical water conservancy projects. Through the preliminary investigation, it is found that steel fiber inserted into concrete can improve compressive and splitting properties better of mass concrete. Therefore, this manuscript analyzes the three types of steel fiber concrete commonly used in the market.

3.In the test of steel fiber concrete in this manuscript, when the volume content is 2%, there will be a little fiber aggregation, forming a "scaffolding effect". The reason is that the steel fiber content is large, and the fiber dispersion will be uneven. This is one of the reasons why this paper uses fuzzy random tool to analyze steel fiber reinforced concrete. It is found that when the fiber content is too big, the "scaffolding effect" weakens the compactness of steel fiber reinforced concrete, so the mechanical performance of concrete can not be significantly increased.

4.In order to ensure that the tests in the paper can be repeated, detailed steps of compressive and splitting tensile tests of steel fiber reinforced concrete are added to the Materials section of the manuscript, including mixing time, curing conditions and specific test protocols. Details of the test procedure can be found in red on the revised manuscript.

5-Conclusion

The conclusion effectively summarizes the key findings of the study, highlighting the influence of different steel fiber types on the mechanical properties of concrete and the development of an enhanced RBF fuzzy neural network model. However, it would benefit from streamlining to eliminate repetitive content, particularly in points (1) and (2). Additionally, incorporating a discussion about the limitations of the current study and suggesting potential directions for future research would provide a more comprehensive and balanced view.

Authors’ response:

Thanks for the reviewers' valuable suggestions. According to the suggestion, the conclusion part is reorganized and refined. In addition, points (1) and (2) are combined and simplified. In particular, point (5) is added to discuss the limitations of the current study and suggest potential directions for future research. The revised conclusions are as follows:

1. The milled steel fibers, owing to their unique shape, enhanced the compressive strength more effectively than the end-hook steel fibers. The performance of end-hook steel fibers showed an initial increase in compressive strength with the augmentation of fiber length and content. Notably, when the fiber length exceeded 50 mm and content surpassed 1%, the benefit of additional fibers diminished. It was observed that milled steel fiber enhanced compressive strength significantly more than the end-hook steel fiber. For the end-hook steel fiber, the compressive strength initially rose with increments in fiber length and content, yet this trend reversed beyond certain thresholds. Specifically, a fiber length greater than 50 mm and a content exceeding 1% resulted in diminishing returns regarding additional fiber enhancements.

2. The inclusion of steel fibers resulted in a pronounced improvement in splitting tensile strength as opposed to compressive strength. This enhancement initially increased with the addition of longer fibers and higher dosages but eventually plateaued and declined. Optimal reinforcement was achieved with fibers measuring 50 mm in length and with lf/Dmax ratio of 1.25 for the given dosage. In comparison, when the fiber content was at 1.5% and the length at 60 mm, the performance gains were less notable. In concrete of lower strength grades, the failure mode often involves the pulling out and rupture of fibers, thus the influence of fiber length and dosage on the mechanical properties is more pronounced.

3. Considering the variables of steel fiber type, length, and content, the compressive and splitting tensile strengths of concrete exhibit considerable variability. To address this, an enhanced RBF fuzzy neural network prediction model was developed by refining the learning strategies for central value and weight adjustments. This model uses the type, length, and content of steel fibers as input variables and outputs predicted values for compressive and splitting tensile strength. The establishment of this RBF fuzzy neural network model offers a novel approach for analysing the uncertain characteristics of the mechanical strengths of secondary steel fiber-reinforced concrete.

4. The RBF fuzzy neural network model was employed to forecast the compressive and splitting tensile strengths of steel fiber-reinforced concrete. The predictive convergence rate of the model is increased by 15%, accuracy was corroborated by a less than 10% deviation between forecasted and actual measurements, which surpasses the precision of conventional formulas. Moreover, the refined algorithm of the model demonstrates high efficiency and accuracy, offering reliable technical backing for the secure implementation of substantial steel fiber concrete ventures, including rock anchor beams and similar constructions.

5. Since the test data of secondary steel fiber reinforced concrete is not comprehensive enough, the RBF fuzzy neural network model has not fully utilized the advantages of prediction. Next, the model will be further optimized by supplementing test data under different engineering conditions and citing literature data under similar conditions.

6. PLOS authors have the option to publish the peer review history of their article (what does this mean?). If published, this will include your full peer review and any attached files.

Authors’ response:

We choose “no” and remain anonymous.

Special thanks to you for your good comments.

We tried our best to improve the manuscript and made some changes in the manuscript. These changes will not influence the content and framework of the paper. And here we did not list the changes but marked in red in revised paper.

We appreciate for Editors/Reviewers’ warm work earnestly, and hope that the correction will meet with approval.

Once again, thank you very much for your comments and suggestions.

Best regards

Correspondence: Song Ling

songlintougao@126.com

Song Ling, Du Chengbin, Yao Yafeng and LI Yongheng

---

## [Decision Letter · Decision Letter 1]

6 Feb 2024

Analysis and prediction of compressive and split-tensile strength of secondary steel fiber reinforced concrete based on RBF fuzzy neural network model

PONE-D-23-39113R1

Dear Dr. Lin,

We’re pleased to inform you that your manuscript has been judged scientifically suitable for publication and will be formally accepted for publication once it meets all outstanding technical requirements.

Kind regards,

Paul Awoyera

Academic Editor

PLOS ONE

Additional Editor Comments (optional):

Reviewers' comments:

Reviewer's Responses to Questions

**Comments to the Author**

1. If the authors have adequately addressed your comments raised in a previous round of review and you feel that this manuscript is now acceptable for publication, you may indicate that here to bypass the “Comments to the Author” section, enter your conflict of interest statement in the “Confidential to Editor” section, and submit your "Accept" recommendation.

Reviewer #2: All comments have been addressed

Reviewer #3: All comments have been addressed

2. Is the manuscript technically sound, and do the data support the conclusions?

Reviewer #2: Yes

Reviewer #3: Yes

3. Has the statistical analysis been performed appropriately and rigorously? 

Reviewer #2: Yes

Reviewer #3: Yes

4. Have the authors made all data underlying the findings in their manuscript fully available?

Reviewer #2: Yes

Reviewer #3: Yes

5. Is the manuscript presented in an intelligible fashion and written in standard English?

Reviewer #2: Yes

Reviewer #3: Yes

6. Review Comments to the Author

Reviewer #2: Note that variables and their superscripts and subscripts are properly formatted and should be contextually consistent.

Reviewer #3: I recommend accepting the paper. . The paper is well-written, informative and relevant to the topic at hand. Therefore, I believe that accepting it will be a positive step towards achieving our goal.

7. PLOS authors have the option to publish the peer review history of their article (what does this mean?). If published, this will include your full peer review and any attached files.

---

## [Editor Report · Acceptance letter]

15 Feb 2024

PONE-D-23-39113R1 

PLOS ONE

Dear Dr. Ling, 

I'm pleased to inform you that your manuscript has been deemed suitable for publication in PLOS ONE. Congratulations! Your manuscript is now being handed over to our production team.

Kind regards, 

on behalf of

Dr. Paul Awoyera 

Academic Editor

PLOS ONE